# Rotation-Invariant Spherical Watermarking via Third-Order $SO(3)$ Representation Coupling

**Pengzhen Chen** [1 2 3]  **Yanwei Liu** [1 3]  **Xiaoyan Gu** [1 2 3]  **Antonios Argyriou** [4]  **Wu Liu** [5]  **Weiping Wang** [1]

## Abstract

Reliable watermarking of panoramic imagery is fundamentally challenged by arbitrary 3D rotations. As panoramas are defined on the sphere, they naturally transform under the action of $SO(3)$, rendering conventional planar representations and augmentation-based robustness strategies inadequate and devoid of theoretical guarantees. To address this, we formulate panoramas as spherical signals and leverage $SO(3)$ representation theory to derive provably rotation-invariant descriptors. While spherical harmonic coefficients transform equivariantly under rotations, the natural invariant constructions are typically limited to zeroth-order statistics which eliminate directional information and severely constrain embedding capacity. In this work, we introduce a principled third-order invariant construction by coupling higher-order $SO(3)$ irreducible representations via tensor products and projecting onto the trivial representation. This yields a spherical invariant bispectrum that preserves phase information while remaining strictly rotation-invariant. Leveraging this property, we embed watermarks into higher-order spherical harmonic coefficients and recover them from invariant bispectral scalars, enabling reliable extraction under arbitrary 3D rotations. We provide a theoretical proof of $SO(3)$ invariance for it and demonstrate experimentally its near-perfect robustness to continuous rotations while maintaining high visual fidelity. Code is available here.

## 1. Introduction

The rapid proliferation of AI-generated content (AIGC) has democratized the synthesis of high-fidelity $360°$ panoramic imagery (Wang et al., 2025). Recent advances, including PanFusion (Zhang et al., 2024a) and text-to-360 models (Zhou et al., 2025b), enable the creation of immersive spherical environments from directly natural language prompts. These technologies are accelerating applications across virtual reality and the Metaverse (Shinde et al., 2023; Zhou et al., 2025a; Tukur et al., 2024), while critically, providing foundational data for World Models (Lu et al., 2025b; Li et al., 2025) and Embodied AI agents (Zheng et al., 2025; Wu et al., 2025). However, this ease of generation simultaneously exacerbates risks regarding copyright infringement and unauthorized content redistribution. Digital watermarking offers a principled mechanism for provenance tracking by embedding imperceptible identity signals into media. Yet, despite substantial progress in deep watermarking for planar images, extending these techniques to panoramic imagery remains fundamentally challenging due to its distinct geometric structure.

A panoramic image is not a signal defined on the Euclidean plane $\mathbb{R}^2$, but rather a function residing on the unit sphere $\mathbb{S}^2$. During consumption, users can freely alter their viewing direction via head-mounted displays, an interaction mathematically modeled by the action of the three-dimensional rotation group, $SO(3)$ (Cohen et al., 2018), on the spherical signal. When represented via standard Equirectangular Projection (ERP), such rotations induce highly non-linear and latitude-dependent distortions, including severe polar stretching and large-scale texture displacement (Makadia & Daniilidis, 2003). Consequently, conventional watermark extraction methods, which rely on pixel-grid alignment or local convolutional consistency in Euclidean space, become inherently unstable under global rotations.

Current deep watermarking frameworks are(Lu et al., 2025a; Hu et al., 2024; Bui et al., 2023; Zhang et al., 2024b; Tancik et al., 2020; Chen et al., 2025a; 2026; Wu et al., 2023; Li et al., 2026) predominantly built upon Convolutional Neural Networks (CNNs) that exploit translational equivariance (Ben Jabra & Ben Farah, 2024). While effective against perturbations like Gaussian noise or JPEG compres-

---

[1]Institute of Information Engineering, Chinese Academy of Sciences [2]School of Cyber Security, University of Chinese Academy of Sciences [3]State Key Laboratory of Cyberspace Security Defense [4]University of Thessaly [5]University of Science and Technology of China. Correspondence to: Yanwei Liu <liuyanwei@iie.ac.cn>, Xiaoyan Gu <guxiaoyan@iie.ac.cn>.

sion, they lack intrinsic robustness to geometric transformations. Prior studies typically resort to data augmentation as a heuristic remedy by injecting distorted samples during training to force the network to memorize attacked variations. However, we argue that this strategy is insufficient for theoretically trusted traceability especially for spherical data. Specifically, since $SO(3)$ is a continuous group describing infinite possible rotations (Esteves et al., 2018), each resulting in a distinct non-linear and complex pixel-space distortion. It is infeasible to exhaustively cover the transformation space via finite augmentation. Robustness obtained in this manner relies on memorization rather than geometric consistency, offering no theoretical guarantees and incurring significant training overhead. This fundamental geometric mismatch between the translational equivariance of planar CNNs and the rotational symmetry of spherical signals indicates that robust panoramic watermarking cannot be achieved through projection-space heuristics. Instead, it necessitates representations that explicitly respect the underlying $SO(3)$ symmetry.

To address these challenges, we propose TRIAD, a theoretically grounded framework for provably robust watermarking that delves into the natural spherical structure of panoramic images. As shown in Figure 1, we model images using a Spherical Harmonics (SH) expansion, which provides a compact and continuous parameterization of the sphere, avoiding the non-uniform sampling artifacts inherent to ERP. In the SH domain, the zeroth-order SH coefficient $c_0$ is strictly rotation-invariant (Kondor, 2025). However, as $c_0$ corresponds to the global average (DC component) of the signal (Sloan, 2008), embedding watermarks in this term would cause noticeable shifts in global luminosity and color, severely degrading perceptual quality. Therefore, we propose a novel embedding strategy based on the third-order spherical bispectrum. Specifically, we leverage higher-order spherical harmonic coefficients to carry the watermark information, which offers substantially greater capacity and improved imperceptibility. To recover this information robustly, we construct a third-order tensor product that couples three $SO(3)$ irreducible representations. From a representation-theoretic perspective, decomposing this tensor product yields a trivial ($l = 0$) component corresponding to the bispectrum. This resulting scalar is mathematically guaranteed to be invariant under $SO(3)$ rotations, enabling reliable recovery of watermark information embedded in rotation-sensitive higher-order coefficients while preserving strict rotation invariance at extraction time.

Our main contributions are summarized as follows:

1. Provable Geometric Robustness. We identify the theoretical limitations of augmentation-based robustness for spherical data and propose a watermarking framework with certified $SO(3)$ invariance, grounded in group representa-

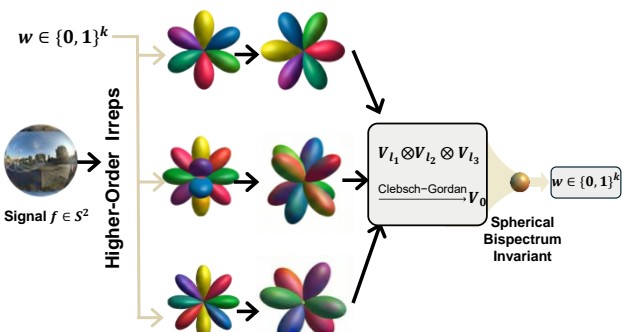

Figure 1. **The grounding theory of TRIAD.** By coupling higher-order spherical harmonics representations and projecting onto the trivial representation, we obtain a scalar invariant (the spherical bispectrum) that retains phase information while remaining invariant to rotations, enabling information embedding in sensitive equivariant coefficients with reliable invariant recovery.

tion theory rather than empirical heuristics.

2. Spherical Bispectrum Invariant. We introduce a novel embedding-to-extraction mechanism based on the spherical bispectrum. By coupling higher-order spherical harmonic coefficients via tensor products, we derive a rotation-invariant scalar that carries messages embedded in higher-order coefficients.

3. TRIAD Framework. We propose TRIAD, an end-to-end framework that seamlessly integrates equivariant operations and invariant extraction. Extensive experiments demonstrate that TRIAD achieves superior robustness against arbitrary $360°$ rotations while maintaining high visual fidelity.

## 2. Related Work

**Robust Watermarking for Panoramic and 3D Data.** Digital watermarking has evolved from traditional frequency-domain methods (DCT/DWT) (Al-Haj, 2007) to deep learning-based frameworks (Zhu et al., 2018; Luo et al., 2020). However, standard CNN-based methods fail to generalize to panoramic images due to the severe geometric distortions inherent in equirectangular projections. While several schemes for $360°$ images have been proposed (Liu et al., 2021), they typically operate in projection space and lack synchronization mechanisms to handle 3D rotations. Similarly, in the 3D data domain, recent works have explored watermarking for meshes (Narendra et al., 2024), point clouds (Zaman et al., 2025), and emerging 3D Gaussian Splatting representations (Chen et al., 2025b), employing techniques such as salient point learning, SVD-based embedding, and neural feature extraction. Despite differences in representation, these approaches share a common reliance on data augmentation during training, which only

provides empirical robustness, and often degrades under unseen transformations, offering no theoretical guarantees. In contrast, our approach exploits the algebraic structure of the rotation group $SO(3)$, enabling mathematically strict rotation-invariance without exhaustive augmentation.

**Spherical CNNs and $SO(3)$-Equivariance.** To handle spherical signals without projection-induced distortion, Cohen et al. (Cohen et al., 2018) introduced Spherical CNNs based on the Fourier theorem on $SO(3)$. Subsequent studies generalized this via $G$-steerable convolutions (Weiler et al., 2018) and tensor field networks (Thomas et al., 2018), establishing the foundation for modern equivariant libraries like e3nn (Geiger & Smidt, 2022). These frameworks leverage Clebsch-Gordan (CG) coefficients (Kondor et al., 2018) to perform tensor products, ensuring that learned feature fields transform predictably under rotations, i.e., equivariantly. While highly effective for discriminative tasks including molecular modeling (Batzner et al., 2022; Kohler et al., 2025) and image processing (Ocampo et al., 2023; Esteves et al., 2018), their application to generative watermarking remains unexplored.

**Higher-Order Invariants and Bispectrum.** Invariant constructions are fundamental for handling geometric transformations, as they enable stable signal representations independent of group actions. A widely adopted approach is the power spectrum (Kazhdan et al., 2003), which achieves invariance by discarding phase information. While effective for recognition tasks (Poulenard et al., 2019), this phase elimination fundamentally limits its capacity of hiding messages. Addressing this requires higher-order statistics. In classical signal processing, the bispectrum (triple correlation) is known to retain phase information (Nikias & Mendel, 1993). Recently, higher-order invariant constructions have been revisited in geometric deep learning, where third-order tensor contractions are employed to capture complex relational structures (Mataigne et al., 2024; Iglesias Martínez et al., 2024; Sanborn & Miolane, 2023). Nevertheless, these studies primarily focus on characterizing fixed physical systems (e.g., atom potentials). Our work is the first to construct a *learnable* spherical bispectrum-based framework specifically for watermarking. By leveraging the phase-preserving property of third-order coupling, we enable high-capacity watermark embedding in rotation-sensitive subspace while extracting from strictly $SO(3)$-invariant bispectral scalars.

## 3. Theoretical Background

This section reviews the mathematical foundations underlying the proposed $SO(3)$-invariant watermarking framework for spherical signals. We briefly introduce rotation group actions, spherical harmonics, and higher-order equivariant constructions, focusing exclusively on the structures essential to our method.

### 3.1. Rotation Group $SO(3)$ and Spherical Signals

Three-dimensional rotations are described by the special orthogonal group $SO(3)$, consisting of all $3 \times 3$ orthogonal matrices with determinant one. For a signal $f(\omega)$ defined on the unit sphere $\mathbb{S}^2$, the action of a rotation $R \in SO(3)$ is defined as:

$$(\mathcal{R}_R f)(\omega) = f(R^{-1}\omega), \tag{1}$$

which corresponds to a rigid rotation of the underlying spherical domain.

Unlike planar images, spherical signals do not admit a global notion of translation. Consequently, the translational equivariance exploited by conventional CNNs has no natural analogue on $\mathbb{S}^2$. When spherical data is represented using planar projections such as equirectangular projection (ERP), rotations in $SO(3)$ induce highly non-uniform, latitude-dependent distortions. As a result, the locality and weight-sharing assumptions underlying standard convolutional kernels are fundamentally violated, motivating the need for representations that explicitly respect rotational symmetry.

### 3.2. Spherical Harmonics

Spherical harmonics (SH) form an orthonormal basis for the space of square-integrable functions on the sphere, $L^2(\mathbb{S}^2)$. Using spherical coordinates $\omega = (\theta, \phi)$, any spherical signal $f(\omega)$ can be expanded as:

$$f(\omega) = \sum_{l=0}^{l_{\max}} \sum_{m=-l}^{l} c_l^m Y_l^m(\omega), \tag{2}$$

where $l$ denotes the frequency degree and $m$ indexes angular variation within each degree. A key property of spherical harmonics is their structured response to rotations. Under a rotation $R$, the SH coefficients transform linearly as:

$$c_l' = D^l(R) c_l, \tag{3}$$

where $c_l \in \mathbb{C}^{2l+1}$ collects the coefficients at degree $l$, and $D^l(R)$ denotes the corresponding Wigner-$D$ matrix. Importantly, coefficients of different degrees do not mix. This block-diagonal transformation law constitutes $SO(3)$-*equivariance* and forms the algebraic backbone of spherical signal processing and equivariant neural networks.

### 3.3. Irreducible Representations and Tensor Products

In equivariant learning frameworks such as e3nn, features are organized as direct sums of irreducible representations (irreps) of $SO(3)$. Each irrep is labeled by its degree $l$ (and parity), with $l = 0$ corresponding to scalars and $l \geq 1$ corresponding to vectors.

The interaction between equivariant features is governed by tensor products. Given two irreducible representations $\mathcal{V}_{l_1}$

and $\mathcal{V}_{l_2}$ of degrees $l_1$ and $l_2$, their tensor product decomposes into a direct sum of irreps:

$$\mathcal{V}_{l_1} \otimes \mathcal{V}_{l_2} \cong \bigoplus_{J=|l_1-l_2|}^{l_1+l_2} \mathcal{V}_J, \quad (4)$$

where the decomposition is mediated by Clebsch–Gordan (CG) coefficients. This operation enables controlled coupling across frequency degrees while preserving equivariance, and crucially allows the construction of invariant features by projecting onto the trivial irrep $\mathcal{V}_0$.

## 4. Methodology

### 4.1. Design Principles

We begin by formalizing the construction of a rotation-invariant watermark derived from a panoramic signal. Let $f : \mathbb{S}^2 \to \mathbb{R}^C$ denote a panoramic image represented in spherical harmonics (SH), with coefficients $c_l^m \in \mathcal{V}_l$, where $\mathcal{V}_l$ denotes the degree-$l$ irreducible representation of $SO(3)$.

The trivial representation $\mathcal{V}_0$ yields rotation-invariant statistics by construction. However it corresponds to global, low-frequency image characteristics, and any perturbation of this component induces perceptually severe artifacts. Intuitively, valid invariant watermarking requires embedding information beyond zeroth-order statistics while retaining the ability to extract it from an invariant quantity. To this end, we exploit third-order correlations among higher-order SH coefficients. Specifically, we consider the tensor product of three irreducible representations:

$$\mathcal{V}_{l_1} \otimes \mathcal{V}_{l_2} \otimes \mathcal{V}_{l_3} = \bigoplus_l \mathcal{V}_l. \quad (5)$$

By projecting the representation onto the trivial subspace $\mathcal{V}_0$, we obtain a scalar invariant derived exclusively from higher-order coefficients, thereby decoupling watermark embedding from invariant extraction. Concretely, the resulting bispectrum invariant $I$ is defined as:

$$I = \sum_{l_1,l_2,l_3} \sum_{m_1,m_2,m_3} C_{l_1 m_1 \, l_2 m_2 \, l_3 m_3}^{0,0} c_{l_1}^{m_1} c_{l_2}^{m_2} c_{l_3}^{m_3}, \quad (6)$$

where $C_{\cdots}^{0,0}$ denotes the Clebsch–Gordan coupling coefficients projecting onto the trivial representation and can be computed via Wigner 3-j symbols:

$$C_{l_1 m_1 l_2 m_2 l_3 m_3}^{0,0} = \sqrt{\frac{(2l_1+1)(2l_2+1)(2l_3+1)}{4\pi}}$$
$$\times \begin{pmatrix} l_1 & l_2 & l_3 \\ 0 & 0 & 0 \end{pmatrix} \begin{pmatrix} l_1 & l_2 & l_3 \\ m_1 & m_2 & m_3 \end{pmatrix}. \quad (7)$$

**Theorem 4.1.** *The bispectrum-based scalar $I$ defined above is invariant under arbitrary $SO(3)$ rotations. Furthermore,*

*perturbations embedded in the higher-order spherical harmonic coefficients induce non-vanishing variations in $I$, satisfying the necessary condition for extraction from this rotation-invariant quantity.*

*Proof.* See Appendix A.1 for the detailed derivation. □

A critical property of Theorem 4.1 is the spectral isolation provided by the SH basis. Due to the orthogonality of spherical harmonics, operations within specific subspaces $\mathcal{V}_{l_1}, \mathcal{V}_{l_2}, \mathcal{V}_{l_3}$ do not introduce interference in unrelated frequency bands. As shown in Figure 2, this enables targeted watermark injection in higher-order components while guaranteeing strict rotation invariance via bispectral projection.

### 4.2. Watermark Embedding

Given an input panorama $x \in \mathbb{R}^{H \times 2H}$ and a watermark vector $w \in \{0,1\}^k$, the encoder embeds watermarks by operating directly in the spherical harmonic (SH) domain. We first lift the input panorama to its spectral representation by computing SH coefficients up to degree $l_{\max}$:

$$c = \{c_l\}_{l=0}^{l_{\max}}, \quad c_l \in \mathcal{V}_l, \quad (8)$$

where each irreducible component transforms equivariantly and independently under rotation.

Watermark embedding is restricted to a selected set of higher-order subspaces:

$$\mathcal{V}_{embed} = \bigoplus_{l \in \mathcal{L}_{embed}} \mathcal{V}_l, \quad l > 0, \quad (9)$$

where $\mathcal{L}_{embed}$ denotes the selected embedding degrees. An $SO(3)$-equivariant backbone $\Phi_{eq}$ is applied to project the raw SH coefficients to selected structured spectral features

$$u = \Phi_{eq}(c), \quad u \in \mathcal{V}_{embed}. \quad (10)$$

Due to the orthogonality of spherical harmonic basis, operations within $\mathcal{V}_{embed}$ do not affect other frequency components.

Watermark information is embedded into $\mathcal{V}_{embed}$ by conditioning the spectral features on the watermark vector from $w$. Specifically, watermark $w$ is mapped to scalar features transforming under the trivial representation $\mathcal{V}_0$, and injected through an $SO(3)$-equivariant interaction with the spectral features. Specifically, we employ a parameterized equivariant tensor product $\text{TP}_{\vartheta_1}$ (detailed in Appendix C.2) to produce a fused update:

$$\Delta u = \text{TP}_{\vartheta_1}(u, w)|_{\mathcal{V}_{embed}}, \quad \text{TP}_{\vartheta_1} : \mathcal{V}_{embed} \otimes \mathcal{V}_0 \to \mathcal{V}_{embed}. \quad (11)$$

By construction, $\Delta u$ lies in the same irreducible subspaces as $u$ and therefore preserves the $SO(3)$ transformation behavior of the spectral representations.

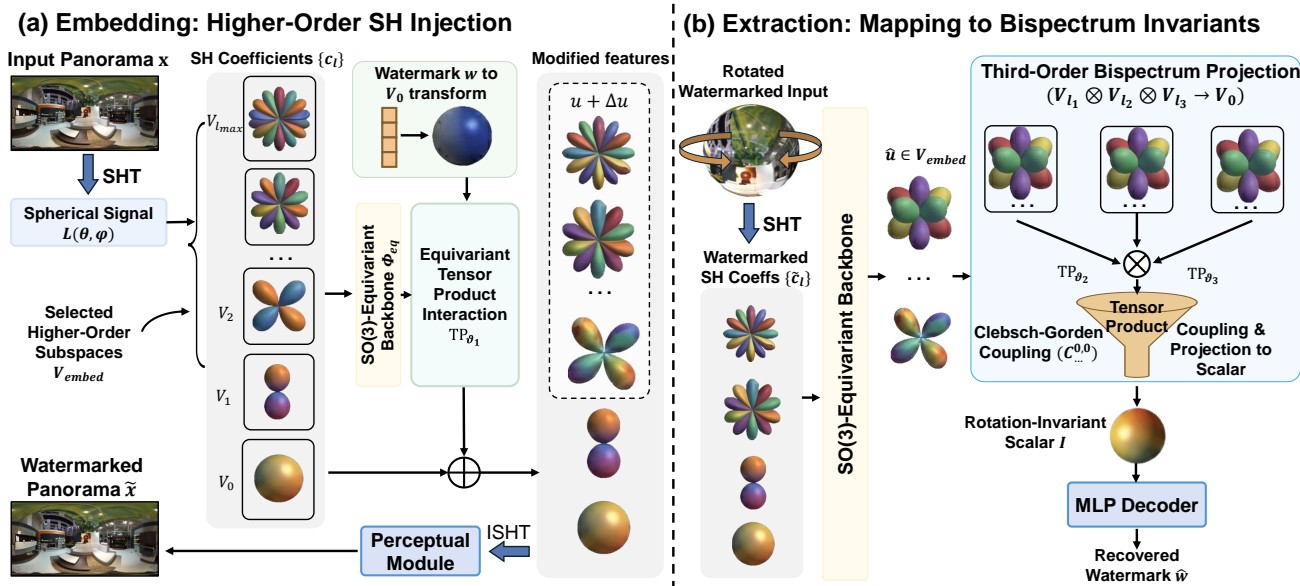

*Figure 2.* **Framework of TRIAD.** (a) Given an input panorama, we first represent it in the spherical harmonics (SH) domain and process the resulting coefficients with an $SO(3)$-equivariant backbone $\Phi_{\text{eq}}$. Watermark information is embedded by modifying selected higher-order SH subspaces, which preserves equivariance and perceptual fidelity under rotations. (b) During extraction, the watermarked SH coefficients are coupled through a third-order tensor product and projected onto the trivial representation using Clebsch–Gordan coefficients. This operation produces a rotation-invariant scalar (the spherical bispectrum), from which the embedded watermark can be reliably extracted regardless of the panorama's orientation.

The modified spectral features are computed by $u + \Delta u$. These features are projected back to the full SH coefficient space and transformed to the spatial domain via the inverse spherical harmonic transform (ISHT), yielding a residual image $\Delta x$. To ensure imperceptibility, the residual is modulated by a perceptual module (Appendix C.1), which is composed of a learnable mask $M_{\text{perc}}(x)$ together with a geometric prior from the structure characteristics of ERP $M_{\text{geo}}$, and added to the original panorama to produce the final watermarked output:

$$\tilde{x} = x + M_{\text{perc}}(x) \odot M_{\text{geo}} \odot \Delta x. \quad (12)$$

### 4.3. Extraction

Given a watermarked panorama $\tilde{x}$, the decoder recovers the embedded watermark by extracting rotation-invariant third-order statistics from its spherical harmonic representation. The panorama is first lifted to the SH domain up to $l_{\max}$:

$$\tilde{c} = \{\tilde{c}_l\}_{l=0}^{l_{\max}}, \quad \tilde{c}_l \in \mathcal{V}_l. \quad (13)$$

These coefficients are projected onto the same embedding space $\mathcal{V}_{embed}$ and processed by an $SO(3)$-equivariant backbone, producing spectral features $\hat{u} \in \mathcal{V}_{embed}$ that contain the watermark-induced perturbations. Subsequently, to construct rotation-invariant descriptors, the decoder computes third-order equivariant statistics of the spectral features. Specifically, we apply two successive parameterized equivariant tensor products:

$$h = \text{TP}_{\vartheta_2}(\hat{u}, \hat{u})|_{\mathcal{V}_{embed}}, \quad z = \text{TP}_{\vartheta_3}(h, \hat{u})|_{\mathcal{V}_0}, \quad (14)$$

where $\text{TP}_{\vartheta_3}$ constrains the output subspace to the trivial representation $\mathcal{V}_0$. This two-stage construction is equivalent to forming a third-order tensor product $\text{TP}(\hat{u}, \hat{u}, \hat{u})$ whose output is projected onto the trivial representation, yielding a learnable bispectrum invariant that preserves the sign of the embedded watermark.

These invariant features are aggregated and mapped through a lightweight MLP network to produce the recovered watermark $\hat{w}$. Since extraction relies exclusively on projections onto the trivial representation $\mathcal{V}_0$, the decoding process is provably invariant to arbitrary 3D rotations.

### 4.4. Loss Functions

The network is trained end-to-end using a weighted combination of image fidelity loss and watermark extraction loss:

$$\mathcal{L}_{total} = \lambda_m \mathcal{L}_{MSE}(x, \tilde{x}) + \lambda_{bce} \mathcal{L}_{BCE}(w, \hat{w}), \quad (15)$$

where $\mathcal{L}_{MSE}$ is Mean Squared Error (MSE) loss to ensure visual quality, and $\mathcal{L}_{BCE}$ is the binary cross-entropy loss for message recovery. $\lambda_m$ and $\lambda_{bce}$ are the weighted factors.

# 5. Experiments

## 5.1. Experimental Setup

**Datasets.** We utilize two publicly available panoramic datasets: panoContext (Zhang et al., 2014) and SUN360 (Xiao et al., 2012). We randomly select 10,000 panoramas for training and 2,000 for testing. All panoramas are resized to $512 \times 1024$ equirectangular format for evaluation. For baseline methods that do not support this resolution, we follow the resolution scaling strategy from TrustMark (Bui et al., 2023) to interpolate the watermark strength (see Appendix C.7), which has been shown to preserve watermarking performance.

**Implementation Details.** The $SO(3)$-equivariant backbone is implemented using e3nn (Geiger & Smidt, 2022), consisting of 2 layers of Gated Blocks operating on identical irreducible representations. Spherical harmonic transform is computed with a cutoff degree of $l_{max} = 16$. We set $\mathcal{L}_{embed} = \{6, 8, 14\}$, corresponding to the subspace $\mathcal{V}_{embed} \in \{128 \times 6e, 128 \times 8e, 64 \times 14e\}$, unless otherwise specified. The network is trained using the Adam optimizer with a learning rate of $10^{-4}$ for 300 epochs. The watermark length is set to $k = 32$ bits. The experiments are conducted on a NVIDIA A100 GPU. The loss weights $\lambda_{BCE}$ is set to 10 and $\lambda_m$ is initially set to 1 and linearly increased to 20 over the second 100 epochs.

**Baselines.** We compare TRIAD against the following open-sourced methods: StegaStamp (Tancik et al., 2020), Sep-Mark (Wu et al., 2023), TrustMark (Bui et al., 2023), Edit-Guard (Zhang et al., 2024b), Robust-Wide (Hu et al., 2024), VINE (Lu et al., 2025a). All methods are evaluated using their released checkpoints unless specified.

**Metrics.** We evaluate the performance using Peak Signal-to-Noise Ratio (PSNR), Structural Similarity (SSIM), and Bit Accuracy. PSNR and SSIM quantify the perceptual quality of watermarked images, and Bit Accuracy measures watermark extraction reliability.

## 5.2. Comparative Performance to Arbitrary $SO(3)$ Rotations against Data Augmentation

To evaluate robustness under unconstrained geometric transformations, we apply random 3D rotations sampled uniformly from $SO(3)$ using random unit quaternions. After rotation, watermark extraction is performed directly, without alignment or inverse transformation.

We compare TRIAD against all baseline methods under this setting. To validate the necessity of theoretical robustness over data augmentation, we additionally report baseline performance with extensive rotation-based augmentation. The augmentation training details are provided in Appendix C.5. As shown in Figure 3, without augmentation, all baseline

methods exhibit severe performance degradation once the input is rotated, with bit accuracy collapsing to near-random guessing. This reflects the fundamental mismatch between spherical rotational symmetries and planar convolutional architectures, whose inductive bias is translational equivariance. In ERP representations, any $SO(3)$ rotation can induce highly non-linear, latitude-dependent distortions that cannot be compensated for by local convolutional filters.

While rotation-based data augmentation partially improves robustness at specific angles encountered during training, the resulting performance remains highly irregular and angle-dependent. Crucially, $SO(3)$ is a continuous group with infinitely many possible elements, and any augmentation strategy can only cover a finite subset of this space. Consequently, models trained with augmentation remain vulnerable to unseen rotations. The oscillatory performance observed across rotation angles indicates that such robustness arises from empirical memorization rather than principled invariance, and does not generalize uniformly over $SO(3)$.

Moreover, aggressive augmentation introduces an additional trade-off between robustness and imperceptibility. To maintain watermark extractability under larger transformation uncertainty, baseline methods must increase embedding strength, which directly degrades perceptual quality, as shown in Appendix Table 7.

In contrast, TRIAD maintains near-perfect bit accuracy across all rotation angles without any data augmentation. This is a direct consequence of the theoretically guaranteed $SO(3)$ invariance of the proposed bispectral construction. The results demonstrate that for continuous and unbounded transformation groups such as $SO(3)$, theoretical invariance is not merely advantageous but necessary for practical and reliable watermarking. Additional visualizations of rotated panoramas are provided in Appendix Figure 8.

## 5.3. General Comparison

**Comparative Robustness against Common Distortions.** We evaluate the robustness across all methods under a range of common image distortions, with results reported in Table 1. Beyond robustness to arbitrary rotations, TRIAD demonstrates strong resilience to diverse perturbations without explicit data augmentation, including JPEG Compression, Gaussian Filter, Gaussian Noise, Median Filter, Resize, Brightness, and Contrast, achieving performance comparable to augmented baselines. In most cases, its performance is comparable to or exceeds that of augmented baselines. Detailed distortion parameters are provided in Appendix C.4.

Notably, several of these robustness properties can be directly attributed to the representation-theoretic structure of the proposed framework. Operations such as resizing and

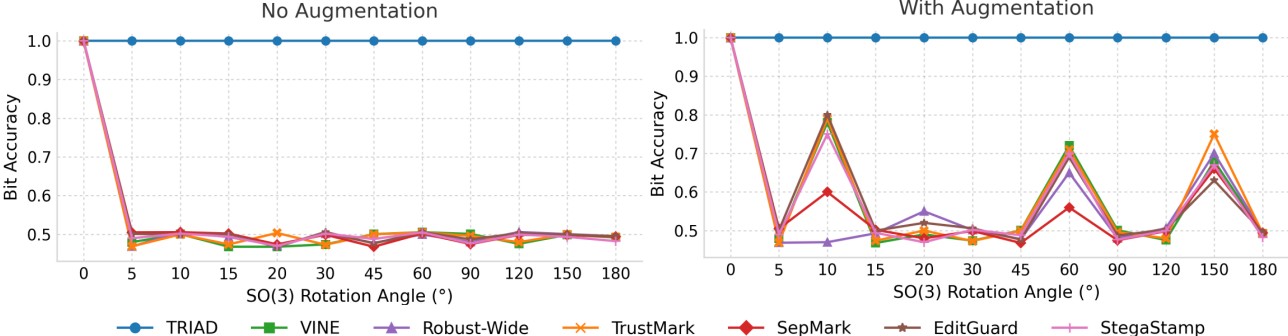

*Figure 3.* Bit accuracy under random $SO(3)$ rotations with increasing rotation angles. We parameterize rotations by their rotation angle, which measures the geodesic distance to the identity rotation on $SO(3)$. For each angle, 1,000 rotation axes are sampled uniformly on $\mathbb{S}^2$, and the average bit accuracy is reported.

*Table 1.* Comparison with baseline methods. The best and the second best results are highlighted in bold and underline, respectively. Mixed denotes the averaged performance over combinations of three randomly selected distortions.

| METHOD | CAPACITY | PSNR↑ | SSIM↑ | GENERAL DISTORTIONS↑ | | | | | | | |
|---|---|---|---|---|---|---|---|---|---|---|---|
| | | | | JPEG | RESIZE | CONTRAST | BRIGHTNESS | GAUSSIAN NOISE | GAUSSIAN BLUR | MEDIAN FILTER | MIXED |
| STEGASTAMP (TANCIK ET AL., 2020) | 100 | 27.96 | 0.8986 | 0.973 | 0.812 | 0.987 | 0.986 | 0.961 | 0.879 | 0.894 | 0.978 |
| SEPMARK (WU ET AL., 2023) | 30 | 35.68 | 0.9799 | 0.985 | 0.864 | 0.988 | 0.984 | 0.978 | 0.987 | 0.969 | 0.977 |
| TRUSTMARK (BUI ET AL., 2023) | 100 | 40.83 | **0.9968** | 0.993 | **1.000** | 0.982 | 0.955 | 0.986 | 0.973 | 0.984 | 0.979 |
| EDITGUARD (ZHANG ET AL., 2024B) | 64 | 36.58 | 0.8865 | 0.957 | 0.634 | 0.966 | 0.941 | 0.935 | 0.513 | 0.546 | 0.679 |
| ROBUST-WIDE (HU ET AL., 2024) | 64 | **41.65** | 0.9921 | 0.997 | 0.998 | 0.973 | **0.990** | 0.989 | 0.999 | **1.000** | **0.992** |
| VINE (LU ET AL., 2025A) | 100 | 36.33 | 0.9865 | **1.000** | **1.000** | 0.994 | 0.976 | **1.000** | 0.951 | 0.965 | 0.986 |
| **TRIAD (OURS)** | 32 | 39.22 | 0.9946 | 0.978 | **1.000** | **1.000** | 0.988 | 0.975 | **1.000** | **1.000** | 0.984 |

Gaussian Filter correspond to isotropic low-pass perturbations in the spherical harmonic domain, inducing smooth, frequency-dependent attenuation of coefficients. Since the proposed watermark is recovered via third-order $SO(3)$-invariant bispectral contractions, which multiplicatively couple multiple coefficients across frequency bands, such spectral attenuation results in a continuous scaling of the invariant response rather than structural destruction, thereby allowing stable extraction. Similarly, regarding additive Gaussian noise, while introducing a bias in higher-order statistics, this isotropic noise primarily induces a magnitude scaling proportional to the signal strength, which preserves the relative geometric configuration of the coupled coefficients. As a result, the bispectral invariant preserves sufficient structure for stable extraction under moderate noise levels. JPEG compression, while non-linear in the spatial domain, predominantly suppresses localized high-frequency content and does not introduce coherent global perturbations aligned with the invariant subspace. This allows reliable recovery of the globally aggregated bispectral features.

In contrast to robustness obtained via data-driven augmentation, these properties arise intrinsically from the algebraic structure of the $SO(3)$-invariant representation. The observed robustness is therefore not incidental, but a direct consequence of embedding watermark information into globally invariant, higher-order spectral statistics. Addi-

tional theoretical analysis is in Appendix A.2.

**Fidelity.** TRIAD achieves high visual fidelity, with PSNR above 39.2 dB and SSIM exceeding 0.99, only marginally lower than TrustMark and Robust-Wide. Visualizations of watermarking patterns are in Appendix Figure 9.

### 5.4. Sensitivity Analysis

**Impact of Embedding Subspace Composition $\mathcal{V}_{embed}$.** We study the effects of spectral composition of the embedding subspace $\mathcal{V}_{embed}$. A fundamental trade-off exists in spherical watermarking: embedding in lower-degree irreducible representations (e.g., $l = 4$) offers inherent geometric stability but induce perceptible low-frequency artifacts, whereas embedding in higher-degree representations (e.g., $l = 16$) ensures improved imperceptibility at the cost of increased vulnerability to high-frequency attenuation during signal processing. We conduct an ablation study on subspace configurations, ranging from single-degree targets to multi-scale combinations. As illustrated in Figure 4, restricting the embedding to low degree ($\mathcal{V}_4$) yields high robustness ($> 99\%$) but compromises visual fidelity. Conversely, exclusively utilizing high degrees ($\mathcal{V}_{16}$) preserves quality but degrades extraction accuracy. Our proposed configuration, $\mathcal{V}_{embed} = \mathcal{V}_6 \oplus \mathcal{V}_8 \oplus \mathcal{V}_{14}$, exploits spectral diversity via direct sums. By distributing the watermark

signal across disjoint irreducible representations, the model effectively balances the objective: it anchor the invariant identity on the robust low-frequency modes while spreading the information payload into the perceptually masked mid-frequency regions. This spectral allocation enables our method to achieve near-perfect recovery while maintaining high perceptual quality (PSNR $\approx 39.2$ dB), significantly outperforming single-degree strategies.

**Analysis of Cutoff Degree $l_{\max}$.** We investigate the trade-off between computational cost, perceptual fidelity, and robustness under varying spherical harmonic cutoff degrees $l_{\max}$. Although increasing $l_{\max}$ theoretically raises the complexity of the Spherical Harmonic Transform, our architecture demonstrates that its impact on inference is limited. Given the same number of selected embedding subspaces $V_{\text{embed}}$, the inference latency remains remarkably stable as $l_{\max}$ increases from 4 to 24. This indicates that the computational bottleneck is dominated by the channel-wise equivariant tensor product operations rather than the spatial-spectral transforms.

Regarding robustness and fidelity, higher cutoff degrees provide access to broader spectral subspaces and thus potentially larger embedding capacity. However, they also introduce a trade-off: higher-degree coefficients correspond to finer spatial details, which are more susceptible to attenuation under lossy compression and aliasing during ERP-to-sphere projection. To examine whether this trend remains stable beyond the default setting, we further extend the ablation to a wider range of cutoff degrees and embedding subspaces. Specifically, we progressively increase $l_{\max}$ from 16 to 28 and enlarge $V_{\text{embed}}$ by incorporating additional higher-degree irreducible subspaces. As summarized in Table 2, the bit accuracy under 3D rotations remains consistently at 100% across all tested configurations, indicating that no obvious numerical instability or abrupt robustness degradation is observed in this range. Meanwhile, PSNR decreases smoothly from 39.22 dB to 37.19 dB as the embedding subspace becomes broader, suggesting that the main effect of using wider spectral subspaces is a gradual reduction in visual fidelity rather than a failure of the rotation-invariant extraction mechanism.

These results further support our choice of $l_{\max} = 16$ as the default configuration. Although larger cutoffs remain robust to 3D rotations in the tested range, they bring limited robustness benefit while gradually sacrificing perceptual quality. Therefore, we adopt $l_{\max} = 16$ together with $V_{\text{embed}} = \{6, 8, 14\}$ as the fidelity–robustness sweet spot, which maximizes the usable embedding capacity within geometrically stable frequency subspaces while avoiding unnecessary fidelity degradation.

**Importance of Bispectrum over Power Spectrum.** Detailed theoretical analysis is provided in the Appendix B.4.

*Table 2.* Ablation over a wider range of cutoff degrees and embedding subspaces. Bit Accuracy is evaluated under 3D rotations.

| $l_{\max}$ | $L_{\text{embed}}$ | PSNR $\uparrow$ | Bit Accuracy $\uparrow$ |
|---|---|---|---|
| 16 | $\{6, 8, 14\}$ | 39.22 | 1.000 |
| 20 | $\{6, 8, 14, 16\}$ | 39.16 | 1.000 |
| 24 | $\{6, 8, 14, 16, 20\}$ | 38.46 | 1.000 |
| 28 | $\{6, 8, 14, 16, 20, 22\}$ | 37.19 | 1.000 |

*Table 3.* Ablation study on invariant projection mechanism. The Power Spectrum fails to support high payload capacities due to phase blindness.

| Projection Mechanism | Order | Capacity (bits) | Bit Acc (%) |
|---|---|---|---|
| Power Spectrum | 2 | 16 | 92.4 |
| Power Spectrum | 2 | 32 | 61.3 |
| **Bispectrum** | **3** | **16** | **100.0** |
| **Bispectrum** | **3** | **32** | **100.0** |
| **Bispectrum** | **3** | **64** | **100.0** |

We evaluate the efficacy of the bispectrum versus the power spectrum as projection mechanisms for mapping higher-order equivariant features to zeroth-order invariant scalars. While both serve as channels to distill rotation-invariant signatures from the embedding space, the bispectrum uniquely retains directional phase information via third-order coupling ($\mathcal{V}_{l_1} \otimes \mathcal{V}_{l_2} \otimes \mathcal{V}_{l_3} \to \mathcal{V}_0$), which is structurally discarded by the second-order Power Spectrum ($\mathcal{V}_{l_1} \otimes \mathcal{V}_{l_2} \to \mathcal{V}_0$). We identify this limitation as "Phase Blindness": the Power Spectrum is a non-injective (many-to-one) mapping, meaning multiple distinct high-order signals can collapse to the same invariant scalar, creating ambiguity that limits watermark capacity. To validate this, we train a variant (*Power-Spec*) in which the bispectral extraction is replaced by power spectrum projection. As shown in Table 6, the *Power-Spec* model fails to converge when the payload exceeds 16 bits. In contrast, the bispectrum preserves phase coupling, allowing distinct signal configurations to be uniquely resolved in the invariant domain. This capability enables TRIAD to scale to higher capacity with near-perfect recovery, validating that third-order statistics are the optimal sufficient statistics for high-capacity invariant watermarking.

## 6. Discussion

Despite the theoretical guarantees of rotation invariance, our framework faces an inherent trade-off between embedding capacity and spectral robustness. As indicated in our analysis (Section 5.4), spherical harmonic components of higher degrees ($l > 16$) are susceptible to attenuation from common distortions such as lossy compression and aliasing artifacts. To prioritize the reliable recovery of the watermark under strictly invariant geometric priors, we constrain the embedding to middle-frequency spectral bands. Conse-

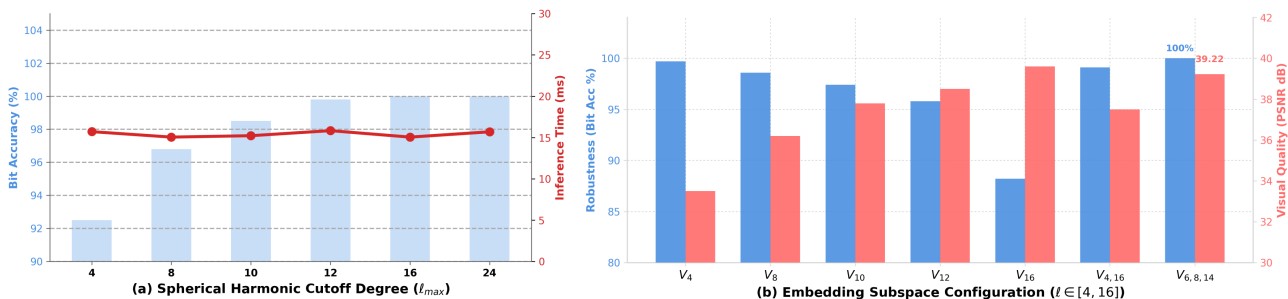

*Figure 4.* (a) Inference time and Bit Accuracy with different spherical harmonic cutoff degree ($l_{max}$). (b) Performance with different settings of embedding subspace $\mathcal{V}_{embed}$. Robustness is evaluated under contrast (0.7x) attack.

quently, while strategies discussed in Appendix C.6 demonstrate potential for capacity enhancement, the effective payload is currently capped (e.g., 64 bits) to ensure stability. Scaling to high-capacity payloads without compromising the robustness of the invariant descriptors remains an open challenge for the spherical watermarking domain. Looking ahead, we will focus on extending the current global invariant framework to handle partial spherical signals by investigating local equivariant symmetries and enhancing the watermark capacity.

## 7. Conclusion

In this work, we present TRIAD, a principled framework for rotation-invariant watermarking of panoramic imagery grounded in the representation theory of $SO(3)$. By modeling panoramas as spherical signals and leveraging third-order representation coupling, we construct bispectral invariants that enable reliable watermark extraction under arbitrary 3D rotations. Unlike augmentation-based approaches, our method provides theoretical guarantees of invariance while preserving high perceptual quality by embedding information exclusively in higher-order spherical harmonic components. Extensive experiments demonstrate that TRIAD achieves near-perfect robustness to continuous $SO(3)$ rotations and strong resilience to common signal distortions. We believe this work highlights the necessity of invariant representations for watermarking on non-Euclidean domains and opens new directions for information hiding based on higher-order group-theoretic invariants.

## Acknowledgements

This work was supported by the Strategic Priority Research Program of the Chinese Academy of Sciences (NO. XDB0690302), and the National Nature Science Foundation of China under Grant 62371450.

## Impact Statement

This work advances digital watermarking for spherical media by introducing a provably rotation-invariant framework grounded in group representation theory. By moving beyond the prevailing reliance on data augmentation, which offers only empirical and bounded robustness, we establish a framework for provably robust watermarking grounded in group representation theory. This theoretical guarantee is critical for the reliable provenance tracking of immersive content in increasingly complex pipelines, such as World Models and the Metaverse, where geometric transformations are intrinsic rather than adversarial. Furthermore, our introduction of the third-order spherical bispectrum as a carrier for information transmission provides a new direction for information hiding. By demonstrating that higher-order spectral invariants can serve as a strictly rotation-invariant domain for embedding and extraction, we provide a mathematically grounded alternative to spatial or frequency-based heuristics. This approach not only solves the immediate challenge of $SO(3)$ robustness but also establishes a generalized methodology for signal processing on non-Euclidean manifolds. We anticipate this will inspire future research into equivariant information carriers for other geometric data types, fostering the development of trustworthy copyright protection mechanisms for the emerging applications in immersive media, 3D vision and embodied AI systems.

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

# A. Theoretical Analysis

## A.1. Proof of Theorem 4.1

*Proof.* Fix arbitrary degrees $(l_1, l_2, l_3)$. Let $R \in SO(3)$ be an arbitrary rotation. Under $R$, the spherical harmonics coefficients transform as

$$c_l^m \;\mapsto\; \sum_{m'=-l}^{l} D_{mm'}^{(l)}(R)\, c_l^{m'},$$

where $D^{(l)}(R)$ denotes the Wigner-$D$ matrix corresponding to the irreducible representation $\mathcal{V}_l$.

Consider the corresponding bispectrum component

$$I_{l_1,l_2,l_3} = \sum_{m_1,m_2,m_3} C^{0,0}_{l_1 m_1\, l_2 m_2\, l_3 m_3}\, c_{l_1}^{m_1} c_{l_2}^{m_2} c_{l_3}^{m_3}.$$

After applying the rotation $R$, the transformed quantity becomes

$$I_{l_1,l_2,l_3}(R) = \sum_{m_1,m_2,m_3} C^{0,0}_{l_1 m_1\, l_2 m_2\, l_3 m_3} \left( \sum_{m_1'} D_{m_1 m_1'}^{(l_1)}(R) c_{l_1}^{m_1'} \right) \left( \sum_{m_2'} D_{m_2 m_2'}^{(l_2)}(R) c_{l_2}^{m_2'} \right)$$
$$\times \left( \sum_{m_3'} D_{m_3 m_3'}^{(l_3)}(R) c_{l_3}^{m_3'} \right).$$

Reordering the summations yields

$$I_{l_1,l_2,l_3}(R) = \sum_{m_1',m_2',m_3'} \left( \sum_{m_1,m_2,m_3} C^{0,0}_{l_1 m_1\, l_2 m_2\, l_3 m_3} D_{m_1 m_1'}^{(l_1)}(R) D_{m_2 m_2'}^{(l_2)}(R) D_{m_3 m_3'}^{(l_3)}(R) \right) c_{l_1}^{m_1'} c_{l_2}^{m_2'} c_{l_3}^{m_3'}.$$

By the invariance property of Clebsch–Gordan coefficients, the contraction of three Wigner-$D$ matrices with $C^{0,0}$ satisfies

$$\sum_{m_1,m_2,m_3} C^{0,0}_{l_1 m_1\, l_2 m_2\, l_3 m_3} D_{m_1 m_1'}^{(l_1)}(R) D_{m_2 m_2'}^{(l_2)}(R) D_{m_3 m_3'}^{(l_3)}(R) = C^{0,0}_{l_1 m_1'\, l_2 m_2'\, l_3 m_3'}.$$

Substituting this identity back, we obtain

$$I_{l_1,l_2,l_3}(R) = I_{l_1,l_2,l_3}.$$

Since the full bispectrum invariant $I$ is obtained by summing $I_{l_1,l_2,l_3}$ over all $(l_1, l_2, l_3)$ and the above argument holds independently for each triple, the complete bispectrum invariant $I$ is invariant under arbitrary $SO(3)$ rotations.

This establishes the rotation invariance of $I$. We next analyze its sensitivity to perturbations in higher-order spherical harmonics coefficients. Consider a small perturbation $\Delta c_l^m$ embedded in higher-order SH coefficients. Due to the linearity of the tensor product and the projection onto the trivial representation, the resulting change in $I$ can be expressed as

$$\Delta I = \sum_{l_1,l_2,l_3} \sum_{m_1,m_2,m_3} C^{0,0}_{l_1 m_1\, l_2 m_2\, l_3 m_3} \Big( \Delta c_{l_1}^{m_1} c_{l_2}^{m_2} c_{l_3}^{m_3} + c_{l_1}^{m_1} \Delta c_{l_2}^{m_2} c_{l_3}^{m_3} + c_{l_1}^{m_1} c_{l_2}^{m_2} \Delta c_{l_3}^{m_3} \Big) + O(\Delta c^2).$$

Hence, $\Delta I$ is non-zero for generic perturbations. This confirms that the embedded information is not lost during the projection to the invariant scalar $I$, but rather manifests as detectable structural changes, providing the discriminative basis for the learnable decoder to recover the watermark. Any information embedded in the higher-order SH coefficients that contributes to $I$ manifests in the rotation-invariant scalar and can, in principle, be extracted given knowledge of the embedding scheme.

$\square$

## A.2. Theoretical Interpretation of Robustness against Common Distortions

The robustness of the proposed watermarking framework to common image distortions arises naturally from the multilinearity, continuity, and global aggregation properties of third-order SO(3)-invariant bispectral representations. Unlike data augmentation, which empirically approximates robustness, these properties follow directly from the algebraic structure of the embedding space, providing principled stability guarantees beyond rotation invariance.

Here, we analyze the theoretical stability of the spherical bispectrum operator. While the proposed TRIAD architecture employs a equivariant backbone prior to bispectrum computation, this theoretical analysis remains highly relevant. As demonstrated in Figure 6, the watermarking signal is basically embedded onto the targeted Spherical Harmonics subspace, which showcases that the backbone $\Phi_{\mathrm{eq}}(\cdot)$ effectively functions as a projection from full coefficients to those of targeted subspace. Therefore, the robustness inherent to the raw SH coefficients below directly extends to our method.

### A.2.1. PRELIMINARIES AND NOTATION

Let $f(\omega)$ be a spherical signal defined on $\mathbb{S}^2$ with spherical harmonic expansion

$$f(\omega) = \sum_{l=0}^{l_{\max}} \sum_{m=-l}^{l} c_l^m Y_l^m(\omega).$$

Following the main paper, we define the third-order SO(3)-invariant bispectrum scalar as

$$I = \sum_{l_1,l_2,l_3} \sum_{m_1,m_2,m_3} C_{l_1 m_1\, l_2 m_2\, l_3 m_3}^{0,0} c_{l_1}^{m_1} c_{l_2}^{m_2} c_{l_3}^{m_3}, \tag{16}$$

where $C_{l_1 m_1\, l_2 m_2\, l_3 m_3}^{0,0}$ denotes the Clebsch–Gordan coefficients corresponding to projection onto the trivial representation. By construction, $I$ is invariant under arbitrary SO(3) rotations.

### A.2.2. STABILITY UNDER ISOTROPIC GAUSSIAN FILTERING

Isotropic Gaussian filtering on the sphere corresponds to convolution with the spherical heat kernel. In the spherical harmonic domain, this induces a frequency-dependent attenuation:

$$\tilde{c}_l^m = g(l)\, c_l^m, \quad g(l) = e^{-\sigma^2 l(l+1)}.$$

Substituting $\tilde{c}_l^m$ into Eq. (16) yields:

$$\tilde{I} = \sum_{l_1,l_2,l_3} \sum_{m_1,m_2,m_3} C_{l_1 m_1\, l_2 m_2\, l_3 m_3}^{0,0} g(l_1)g(l_2)g(l_3)\, c_{l_1}^{m_1} c_{l_2}^{m_2} c_{l_3}^{m_3} \tag{17}$$

$$= \sum_{l_1,l_2,l_3} g(l_1)g(l_2)g(l_3) \sum_{m_1,m_2,m_3} C_{l_1 m_1\, l_2 m_2\, l_3 m_3}^{0,0} c_{l_1}^{m_1} c_{l_2}^{m_2} c_{l_3}^{m_3}. \tag{18}$$

Equivalently, if $I_{l_1,l_2,l_3}$ denotes the bispectral contribution of the triplet $(l_1, l_2, l_3)$, then

$$\tilde{I} = \sum_{l_1,l_2,l_3} g(l_1)g(l_2)g(l_3) I_{l_1,l_2,l_3}. \tag{19}$$

This shows that isotropic blur acts on the bispectrum through a structured and degree-dependent attenuation of frequency-triplet responses. Importantly, the coupling pattern induced by the Clebsch–Gordan projection is preserved: blur rescales each valid triplet contribution without mixing unrelated irreducible subspaces. For a fixed bandwidth $l_{\max}$, the induced variation is bounded by

$$|\tilde{I} - I| \leq \sum_{l_1,l_2,l_3 \leq l_{\max}} |g(l_1)g(l_2)g(l_3) - 1|\, |I_{l_1,l_2,l_3}|. \tag{20}$$

Therefore, under moderate blur, the bispectral descriptor undergoes controlled attenuation rather than unstructured distortion. This structured response explains why the learned extractor can retain reliable recovery when sufficient low- and mid-frequency bispectral components remain.

A.2.3. EFFECT OF RESIZING AND LOW-PASS PERTURBATIONS

Resizing with anti-aliasing can be approximated as a low-pass operation in the spherical harmonic domain:

$$\tilde{c}_l^m = \begin{cases} c_l^m, & l \leq l_c, \\ 0, & l > l_c. \end{cases} \tag{21}$$

The corresponding bispectrum is then restricted to frequency triplets whose degrees remain below the cutoff:

$$\tilde{I} = \sum_{l_1,l_2,l_3 \leq l_c} \sum_{m_1,m_2,m_3} C_{l_1 m_1 l_2 m_2 l_3 m_3}^{0,0} c_{l_1}^{m_1} c_{l_2}^{m_2} c_{l_3}^{m_3}. \tag{22}$$

Thus, low-pass perturbations remove bispectral interactions involving truncated high-frequency components, while preserving the interactions among retained low- and mid-frequency irreducible subspaces. For natural images, spectral energy is typically concentrated in low- and mid-frequency bands, whereas very high-frequency components carry comparatively weaker energy and are more sensitive to sampling artifacts. Consequently, moderate resizing primarily suppresses high-frequency triplets while leaving a substantial portion of signal-dependent bispectral structure intact.

To make this statement precise, let

$$\mathcal{T}_c = \{(l_1, l_2, l_3) : l_1, l_2, l_3 \leq l_c\} \tag{23}$$

denote the retained set of triplets. The amount of preserved bispectral information can be characterized by the retained bispectral energy ratio

$$\rho(l_c) = \frac{\sum_{(l_1,l_2,l_3) \in \mathcal{T}_c} |I_{l_1,l_2,l_3}|^2}{\sum_{l_1,l_2,l_3 \leq l_{\max}} |I_{l_1,l_2,l_3}|^2}. \tag{24}$$

When $\rho(l_c)$ remains sufficiently large, the invariant descriptor preserves discriminative, input-dependent structure after low-pass filtering. In this sense, the representation remains informative rather than degenerating into an input-independent or nearly constant descriptor. This formulation clarifies the role of low-/mid-frequency bispectral couplings in maintaining robust extraction under resizing.

A.2.4. ROBUSTNESS TO ADDITIVE GAUSSIAN NOISE.

We model additive noise in the spherical harmonic domain as:

$$\tilde{c}_l^m = c_l^m + \epsilon_l^m, \quad \epsilon_l^m \sim \mathcal{N}(0, \sigma^2).$$

To evaluate the stability of the watermarking signal, we examine the expectation of the third-order bispectrum invariant under this perturbation. Substituting the noisy coefficients into the tensor contraction (Eq. (16)) and taking the expectation, we observe that while the first-order noise terms vanish (due to $\mathbb{E}[\epsilon] = 0$), the second-order interactions introduce a non-zero term:

$$\mathbb{E}[\tilde{I}] = I + \sum C_{l_1 l_2 l_3}^{0,0} \left( c_{l_1} \cdot \mathbb{E}[\epsilon_{l_2} \epsilon_{l_3}] + \dots \right) + \mathbb{E}[\epsilon^3]$$

Since the noise is isotropic, the quadratic term $\mathbb{E}[\epsilon^2]$ is proportional to the noise variance $\sigma^2$. Consequently, the expected value of the noisy invariant takes the form:

$$\mathbb{E}[\tilde{I}] \approx I + \beta \cdot c \cdot \sigma^2 \approx I(1 + \lambda \sigma^2)$$

where $\lambda$ is a scalar factor derived from the coupling constants.

This result indicates that while the estimator is mathematically biased (contradicting a zero-bias assumption), the bias is structure-preserving. Specifically, the noise acts primarily as a magnitude scaling factor proportional to the signal strength $c$, rather than an additive shift that disrupts the feature's sign or relative orientation. This geometric property explains the experimental robustness without explicit noise augmentation. Since the perturbation scales the feature vector but preserves its directionality and relative ordering, it does not push the embedding across the decision boundary of the MLP decoder. Thus, the watermark remains recoverable even when the feature magnitude is modulated by noise interference. (Note: While the real-valued constraint of images imposes conjugate symmetry on $\epsilon$, implying correlations between $\epsilon^m$ and $\epsilon^{-m}$, this strictly affects the magnitude of the scalar $\lambda$ but does not alter the fundamental conclusion that the distortion manifests as a signal-dependent scaling.)

*Table 4.* Isolated analysis of bispectral descriptors under common distortions. Cosine similarity is computed between the bispectrum before and after applying each distortion.

| Distortion | Strength | Cosine Similarity ↑ |
|---|---|---|
| Gaussian Blur | $\sigma = 3$ | 0.997849 |
| Resize | $0.5\times$ | 0.999972 |
| Gaussian Noise | std. $= 0.05$ | 0.999965 |
| Combined | – | 0.997458 |

### A.2.5. JPEG COMPRESSION AND LOCALIZED DISTORTIONS

JPEG compression is non-linear in the spatial domain and difficult to model exactly in spherical harmonics. However, its primary effect consists of localized block-wise quantization and high-frequency suppression. Since bispectral invariants aggregate global spectral interactions through Clebsch–Gordan contractions, such localized distortions do not coherently align with the invariant subspace. Consequently, the global bispectral response remains stable, consistent with our empirical observations.

### A.2.6. ISOLATED QUANTITATIVE ANALYSIS OF BISPECTRAL DESCRIPTORS

To further validate the stability properties analyzed above, we evaluate the bispectral descriptor in isolation under representative distortions. This experiment directly measures the change of the invariant representation itself, before the learning-based watermark decoder.

For each clean panorama $x$, we compute its bispectral descriptor $B(x)$ from the spherical harmonic coefficients. Given a distorted image $\hat{x}$, we compute $B(\hat{x})$ using the same cutoff degree and frequency-triplet configuration. We then measure the cosine similarity between the two descriptors:

$$S_{\cos} = \frac{\langle B(\hat{x}), B(x) \rangle}{\|B(\hat{x})\|_2 \|B(x)\|_2}. \tag{25}$$

A higher cosine similarity indicates that the bispectral descriptor preserves its direction in the invariant feature space after distortion.

As shown in Table 4, the bispectral descriptor remains highly consistent under isolated distortions, with cosine similarity above 0.997 even under the combined setting. These results quantitatively support the stability analysis above: common distortions perturb the bispectrum in a structured and smooth manner, while largely preserving the direction of the invariant descriptor.

## B. More Analysis

### B.1. Overhead Evaluation

We evaluate the encoding, decoding and total time cost and GPU memory usage of watermarking methods on an NVIDIA A100 GPU. The results are averaged over 1,000 images. As shown in Table 5, our method demonstrates a comparatively low cost both in inference time and in GPU usage.

### B.2. Spectral Precision and Orthogonal Embedding

To verify the disentanglement capabilities of our encoder, we examine the distribution of watermark energy across the spherical harmonic degrees. We decompose the image signal into its spectral components and compute the power spectrum $P(l)$ for each degree $l$:

$$P(l) = \sum_{m=-l}^{l} \|c_l^m\|^2$$

where $c_l^m$ denotes the spherical harmonic coefficients.

Ideally, the embedding scheme should modulate only the specific degrees allocated for the payload, ensuring that the information is strictly confined to the target subspaces $\mathcal{V}_{embed}$ without leaking into or corrupting adjacent coefficients.

*Table 5.* Comparison of watermarking methods based on running time per single image and GPU memory usage. The results are averaged over 1,000 images.

| Method | Encoding Time (s) | Decoding Time (s) | Total Time (s) | Memory (GB) |
|---|---|---|---|---|
| StegaStamp (Tancik et al., 2020) | 0.0354 | 0.0318 | 0.0672 | 2.0 |
| SepMark (Wu et al., 2023) | 0.0053 | 0.0055 | 0.0108 | 0.9 |
| TrustMark (Bui et al., 2023) | 0.0197 | 0.0147 | 0.0344 | 0.6 |
| EditGuard (Zhang et al., 2024b) | 0.1659 | 0.0773 | 0.2432 | 1.7 |
| Robust-Wide (Hu et al., 2024) | 0.0111 | 0.0157 | 0.0268 | 3.0 |
| VINE (Lu et al., 2025a) | 0.0583 | 0.0103 | 0.0686 | 4.9 |
| **TRIAD (Ours)** | **0.0081** | **0.0072** | **0.0153** | **0.8** |

Figure 6 presents the spectral energy difference between the cover and watermarked images. These results demonstrate the high precision of our modulation mechanism, where energy variations are only observed exclusively at the pre-defined embedding degrees from $\mathcal{V}_{embed}$. This confirms that the embedding strategy correctly map the latent watermark code onto the intended geometric features. While for the non-targeted degrees, the coefficient norms remain virtually identical to those of the original image (relative change $\approx 0$). This indicates that our method achieves near-perfect orthogonality of spectral watermark injection. The absence of unintended perturbations in non-watermarked degrees validates the functionality of the extraction process and preserves the image quality by leaving the vast majority of the frequency components unchanged, which aligns perfectly with our design objective of localized and disentangled feature modulation.

### B.3. Verification of Rotational Invariance

To empirically validate the theoretical invariance of our bispectrum-based architecture, we conduct a rigorous stability analysis on the latent feature space. Specifically, given an input $x$ and a randomly rotated counterpart $x_{rot} = g \cdot x$ (where $g \in SO(3)$), we extract their respective tensor product feature vectors $\mathbf{z}$ and $\mathbf{z}_{rot}$ and analyze their correspondence. As illustrated in Fig 7, the Scatter Correlation analysis reveals a near-perfect linear relationship between $\mathbf{z}$ and $\mathbf{z}_{rot}$. All feature points densely cluster around the diagonal identity line ($y = x$), demonstrating that the feature magnitude remains consistent regardless of the input's geometric orientation. The Pearson correlation coefficient is close to $1.0$, indicating strong linear dependence. Furthermore, the overlay plot offers a microscopic view of the feature topology. We observe that the red dashed line (representing $\mathbf{z}_{rot}$) explicitly tracks the blue solid line (representing $\mathbf{z}$) across feature dimensions. The overlapping waveforms indicate that the bispectrum contraction mechanism successfully eliminates the perturbations induced by rotation. These observations align with our theoretical derivation. By computing the third-order invariants (bispectrum) via the contraction $\mathbf{TP}(\mathrm{TP}(u,u),u) \to \mathcal{V}_l$, the model effectively cancels out the Wigner-D matrices induced by rotation. This confirms that our decoder has learned a robust $SO(3)$-invariant representation, ensuring reliable watermark extraction even under extreme geometric distortions.

### B.4. From Power Spectrum to Bispectrum

Achieving robustness to arbitrary 3D rotations requires extracting watermark information from representations that are strictly invariant under $SO(3)$. Intuitively, we observe that it can be achieved by constructing invariants from spherical harmonic (SH) coefficients via tensor contractions that project onto the trivial representation.

The simplest such invariant is the power spectrum,

$$P_l = \sum_{m=-l}^{l} |c_l^m|^2 = \|\mathbf{c}_l\|^2, \tag{26}$$

which can be interpreted as a second-order contraction of $f \otimes f$ onto the scalar ($0e$) irreducible representation. Since Wigner-$D$ matrices are unitary, $P_l$ is exactly invariant under arbitrary rotations. However, this construction discards all relative phase information between SH coefficients. As a result, the mapping from signals to power spectra is highly non-injective: structurally distinct signals may share identical power spectra. For watermarking, this phase blindness fundamentally limits both embedding capacity and reconstructability.

To overcome this limitation, we turn to higher-order invariant constructions. In particular, third-order tensor products

allow the preservation of phase coupling while maintaining exact $SO(3)$ invariance. This leads naturally to the bispectrum invariant, which is obtained by contracting three SH coefficients via Clebsch–Gordan (CG) coefficients corresponding to projection onto the trivial representation:

$$I = \sum_{l_1, l_2, l_3} \sum_{m_1, m_2, m_3} C^{0,0}_{l_1 m_1 \, l_2 m_2 \, l_3 m_3} c^{m_1}_{l_1} c^{m_2}_{l_2} c^{m_3}_{l_3}. \tag{27}$$

By construction, $I$ is a scalar invariant under $SO(3)$.

Unlike the power spectrum, the bispectrum retains closed-loop phase coupling across frequencies, resolving much of the non-injectivity inherent to second-order invariants. Classical results in harmonic analysis show that, under mild conditions, bispectral invariants uniquely characterize a signal up to a global rotation. Consequently, embedding watermark information into the bispectrum invariant space achieves exact rotational robustness while substantially increasing representational capacity.

In our framework, this invariant arises naturally from the quadratic tensor product of equivariant features, followed by projection onto the $0e$ subspace. This representation-theoretic construction ensures that robustness to arbitrary rotations is achieved by design, rather than through data augmentation, while preserving the structural information necessary for reliable watermark decoding.

## C. More Details

### C.1. Perceptual Module Detail

The perceptual module modulates the watermark residual using two complementary components to ensure imperceptibility. First, a Learnable Content Mask is generated by a lightweight CNN (consisting of three convolutional layers with SiLU activations and a final Sigmoid) to adaptively hide information in high-texture regions.

Second, we apply a fixed Geometric Prior $M_{geo}(\theta) = \sin(\theta)$ to counteract the non-uniform sampling of Equirectangular Projection (ERP). Since the panoramic image is an Equirectangular Projection (ERP) of the sphere, pixels do not represent equal areas. The sampling density increases significantly near the poles ($\theta \to 0, \pi$), meaning modifications in these regions are spatially magnified when viewed on the sphere. To counteract this distortion, we introduce a fixed geometric prior $M_{geo}$ derived from the spherical area element $dA = \sin\theta d\theta d\phi$. For an image of height $H$, the weight for row $i$ (corresponding to polar angle $\theta_i$) is calculated as:

$$M_{geo}(i) = \sin(\theta_i), \quad \text{where } \theta_i = \frac{i}{H}\pi \tag{28}$$

This sinusoidal weighting suppresses watermark strength near the poles while allowing full strength at the equator, preventing polar artifacts.

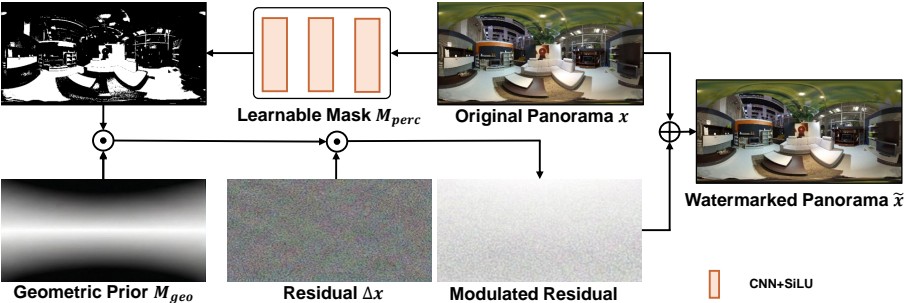

*Figure 5.* **Perceptual module.** The learnable mask $M_{perc}$ is three blocks of Convolution networks followed by an activation.

### C.2. Network Architecture Details

We provide the detailed specifications of the core modules.

1. Equivariant Tensor Product Interaction $\text{TP}_\vartheta$. To enable information mixing within the spherical harmonic domain, we employ a `FullyConnectedTensorProduct` from `e3nn` (Geiger & Smidt, 2022) library. We define the parameterized equivariant tensor product, denoted as $\text{TP}_\vartheta(\cdot, \cdot)|_{V_{out}}$, which couples two input representations using fixed Clebsch-Gordan coefficients followed by learnable path weights to map the result to a specified output subspace $V_{out}$.

2. Watermark $w$ Mapping. We apply 3 three layers of linear and activation block to project $w$ from $k$ to $D_w = 512$ dimensions.

3. $SO(3)$-Equivariant Backbone. The backbone consists of stacked Gated Blocks, which are essential for introducing non-linearity to geometric tensors without breaking equivariance. Each block performs the following operations:

- Linear Projection: An equivariant linear layer.

- Gating Mechanism: To activate higher-order tensors (which do not have a rotation-invariant sign), the features are separated into scalars and gated tensors.

  - Scalars ($0e$): Activated directly using the SiLU function.
  - Higher-order Tensors ($l > 0$): Modulated element-wise by a learned scalar gate (activated via Sigmoid to $[0, 1]$).

This structure ensures that the magnitude of directional features is non-linearly transformed while their orientation remains equivariant.

4. MLP Decoder. The decoder maps the extracted rotation-invariant bispectral scalars to the binary watermark vector. It follows a standard Multi-Layer Perceptron structure of: A sequence of Linear layers ($dim \rightarrow 256 \rightarrow 128$) with SiLU activations and a final Linear projection maps the features to the target watermark dimension.

### C.3. More Ablation Analysis

We validate the necessity of the proposed Perceptual Module by removing it from the embedding pipeline (w/o Perceptual Module). As shown in Table 6, disabling the module leads to a sharp decline in visual quality, with PSNR dropping from 39.22 dB to 34.15 dB. Without the learnable mask to suppress perturbations in smooth regions (e.g., sky or walls), the watermark becomes visually intrusive, confirming that the perceptual module helps to maintain high fidelity.

We further analyze the optimal dimension $D_w$ for the watermark projection MLP. Our method set $D_w = 512$. Reducing the dimension to 256 restricts the representation capacity, causing the Bit Accuracy to fall to 96.8%, as the bottleneck limits the effective encoding of the watermark signal. Meanwhile, increasing the dimension to 1024 introduces no gain in both bit accuracy and visual fidelity. This confirms that our design choice ($D_w = 320$), which confirms our setting is efficient yet effective.

*Table 6.* **Ablation study on network components and hyper-parameters.** We investigate the contribution of the Perceptual Module and the impact of the watermark expansion dimension ($D_w$). The default setting (Ours) uses $D_w = 512$ with the Perceptual Module enabled.

| Method / Variant | PSNR (dB) ↑ | Bit Acc (%) ↑ |
|---|---|---|
| **TRIAD (Ours)** | **39.22** | **100.0** |
| *Component Analysis* | | |
| w/o Perceptual Module | 35.15 | 100.0 |
| *Watermark Expansion Dimension ($D_w$)* | | |
| $D_w = 256$ | 39.45 | 96.8 |
| $D_w = 1024$ | 38.60 | 100.0 |

### C.4. Distortions

In our method, we apply a set of commonly used noise perturbations and image transformations to evaluate robustness and performance under degraded visual conditions. The detailed parameter settings are summarized as follows:

- **JPEG Compression**: quality factor = 60.

- **Resize**: scaling factor = 0.5.

- **Gaussian Blur**: kernel size = 1, $\sigma = 3$.

- **Gaussian Noise**: mean = 0, standard deviation = 0.05.

- **Median Filter**: kernel size = 3.

- **Brightness Adjustment**: range = [0.7, 1.3].

- **Contrast Adjustment**: range = [0.7, 1.3].

## C.5. Additional Baseline Rotation Augmentation Training Details

Unlike the continuous rotation invariance guaranteed by our method, baseline models lack intrinsic geometric priors and must rely on seeing specific transformations during training. To simulate this, we randomly select a fixed set of $K = 10$ discrete rotations from the continuous group $SO(3)$ as anchor augmentations. We limit the set size to $K = 10$ because $SO(3)$ rotations induce drastic, non-linear geometric deformations in the equirectangular domain. Attempting to train on a dense or continuous sampling of the rotation manifold introduces excessive distributional variance, which we found empirically to prevent the baseline models from converging. Thus, this limited set serves as a tractable representative sample of the global rotation space to ensure fair comparison. We fine-tune all baseline methods on the PanoContext and SUN360 datasets to evaluate their empirical robustness. Specifically, for each training step, a rotation is stochastically selected from this pre-defined set and applied to the image after watermark embedding. All models are initialized from their officially released checkpoints and trained for 100 epochs using the AdamW optimizer with a learning rate of $1 \times 10^{-4}$.

We evaluate the performance of these augmented baselines in Table 7. As observed, visual fidelity drops significantly due to the inherent trade-off between imperceptibility and robustness in non-equivariant models. Crucially, even when trained and evaluated on this highly restricted subset—which is far simpler than real-world continuous rotation—their performance remains unsatisfactory. This failure to generalize even to a finite set further demonstrates the necessity of theoretical invariance over data augmentation for $SO(3)$ robustness.

*Table 7.* Performance of baseline methods on PSNR and Average Bit Accuracy on panoramas rotated from the pre-defined anchor rotation set. The results are averaged over 1,000 images and angles from the set.

| Method | PSNR (dB) | Average Bit Accuracy (%) |
|---|---|---|
| StegaStamp (Tancik et al., 2020) | 25.15 | 73.38% |
| SepMark (Wu et al., 2023) | 31.42 | 76.65% |
| TrustMark (Bui et al., 2023) | 35.50 | 78.15% |
| EditGuard (Zhang et al., 2024b) | 32.18 | 77.14% |
| Robust-Wide (Hu et al., 2024) | 37.10 | 82.23% |
| VINE (Lu et al., 2025a) | 31.85 | 79.40% |

## C.6. Group-wise Extension for Payload Enhancement

In the main paper, watermark embedding is performed in the embedding space $\mathcal{V}_{embed}$. To increase capacity, we introduce a group-wise extension that partitions $\mathcal{V}_{embed}$ into multiple independent subspaces.

Concretely, the embedding space is composed of irreducible representations with multiplicity:

$$\mathcal{V}_{embed} = \bigoplus_{l \in \mathcal{L}_{embed}} (m_l \times \mathcal{V}_l), \tag{29}$$

where $m_l$ denotes the multiplicity of degree-$l$ irreducible representations. We evenly partition the multiplicity dimension into $G$ groups, yielding:

$$m_l \times \mathcal{V}_l = \bigoplus_{g=1}^{G} \left( \tfrac{m_l}{G} \times \mathcal{V}_l \right), \quad \forall l \in \mathcal{L}_{embed}. \tag{30}$$

The watermark vector $w$ is partitioned into $\{w^{(g)}\}_{g=1}^{G}$. For each group $g$, watermark information is injected only into the corresponding multiplicity slice:

$$\Delta u^{(g)} = \text{TP}_{\vartheta_1}^{(g)}(u^{(g)}, w^{(g)})|_{\left(\frac{m_l}{G} \times \mathcal{V}_l\right)}, \quad \text{TP}_{\vartheta_1}^{(g)} : \left(\frac{m_l}{G} \times \mathcal{V}_l\right) \otimes V_0 \to \left(\frac{m_l}{G} \times \mathcal{V}_l\right). \tag{31}$$

The full modified representation is obtained by concatenating all group-wise updates along the multiplicity dimension.

During extraction, invariant statistics are computed independently within each group:

$$z^{(g)} = \text{TP}\Big(\hat{u}^{(g)}, \hat{u}^{(g)}, \hat{u}^{(g)}\Big), \tag{32}$$

followed by concatenation of the group-wise decoded outputs.

Importantly, groups are formed by splitting the multiplicity dimension of each irreducible representation, rather than by mixing different degrees $l$, which preserves orthogonality and $SO(3)$-equivariance within each group.

### C.7. Resolution Scaling

Algorithm 1 is adapted from the scaling method proposed in TrustMark (Bui et al., 2023). It allows a watermark model trained at a fixed resolution to be applied to images of arbitrary resolutions without performance degradation.

---

**Algorithm 1** Resolution scaling - watermark embedding on arbitrary resolution images

---

**Input:** Original image $\mathbf{x}$, [binary watermark vector $\mathbf{w}$]
**Output:** Watermarked image $\mathbf{y}$
**Data:** Embedding network $E$ trained on resolution $m \times n$
1: $H, W \leftarrow \text{height}(\mathbf{x}), \text{width}(\mathbf{x})$
2: $\mathbf{x} \leftarrow \mathbf{x}/127.5 - 1$                   // Normalize to range $[-1, 1]$
3: $\tilde{\mathbf{x}} \leftarrow \text{interpolate}(\mathbf{x}, (m, n))$
4: $\mathbf{r} \leftarrow E(\tilde{\mathbf{x}}, \mathbf{w}) - \tilde{\mathbf{x}}$                     // Residual image
5: $\mathbf{r} \leftarrow \text{interpolate}(\mathbf{r}, (H, W))$
6: $\mathbf{y} \leftarrow \text{clamp}(\mathbf{x} + \mathbf{r}, -1, 1)$
7: $\mathbf{y} \leftarrow \mathbf{y} \times 127.5 + 127.5$

---

# D. More Results

### D.1. Visualizations of $SO(3)$ Rotations

See Fig 8.

### D.2. Visualizations of Comparative Embedding

See Fig 9.

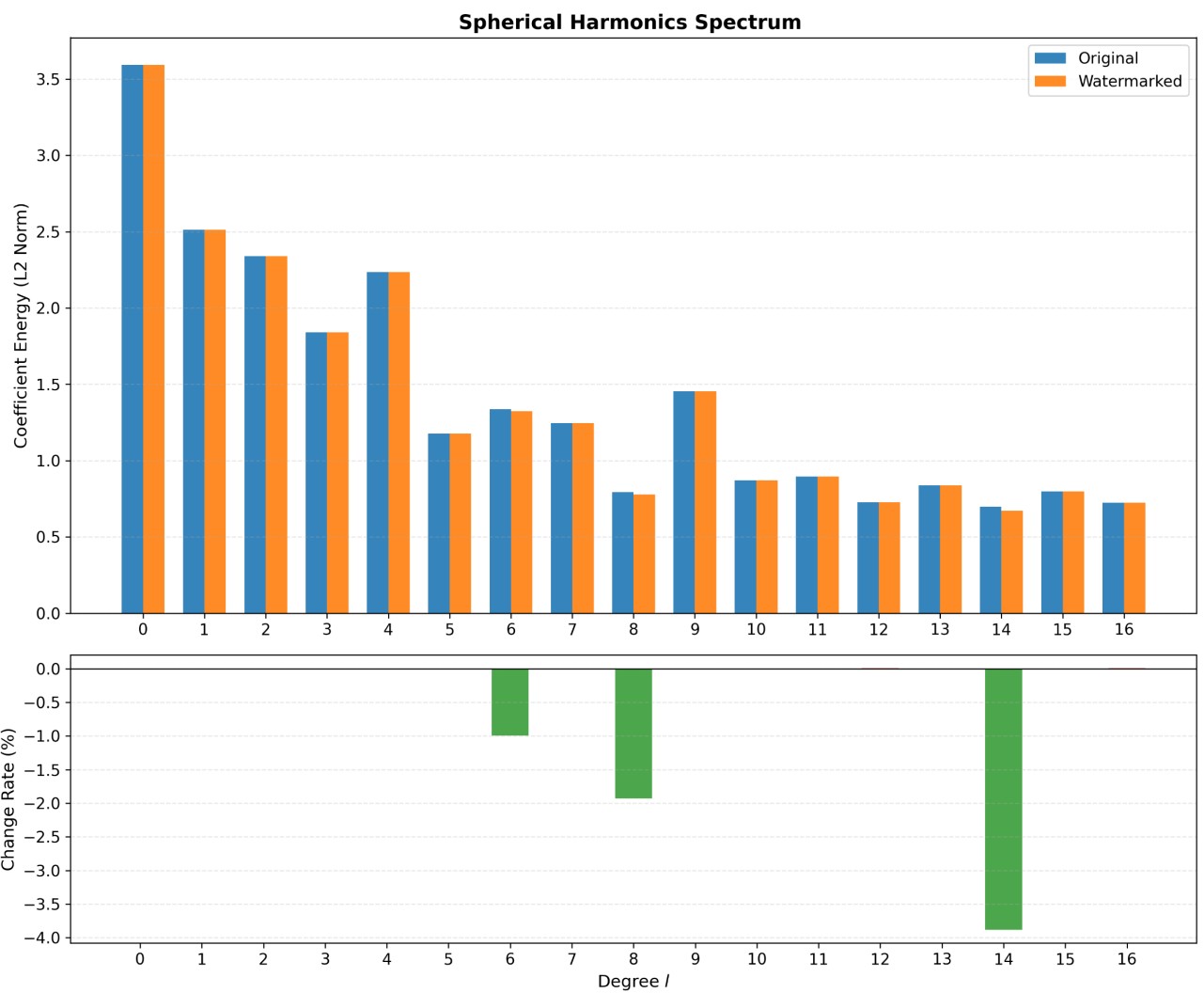

*Figure 6.* Analysis of Spectral Targeting Precision in Spherical Harmonics. We compare the aggregated energy spectrum (L2 norm of coefficients) across degrees $l = 0$ to $l_{max}$ for the original (blue) and watermarked (orange) images. (Top) The spectral profiles exhibit high congruence, indicating that the watermarking process preserves the natural frequency distribution of the cover image. (Bottom) The relative change rate shows distinct, isolated modulations only at the specific degrees targeted for watermark embedding (e.g., $l \in \mathcal{L}_{target}$) and confirms that non-targeted degrees ($l \notin \mathcal{L}_{target}$) remain virtually perturbed (near-zero deviation). This validates the orthogonality of our embedding mechanism, demonstrating that the watermark is accurately confined to the designated subspaces without spectral leakage, thereby preserving the fidelity of unmodulated components.

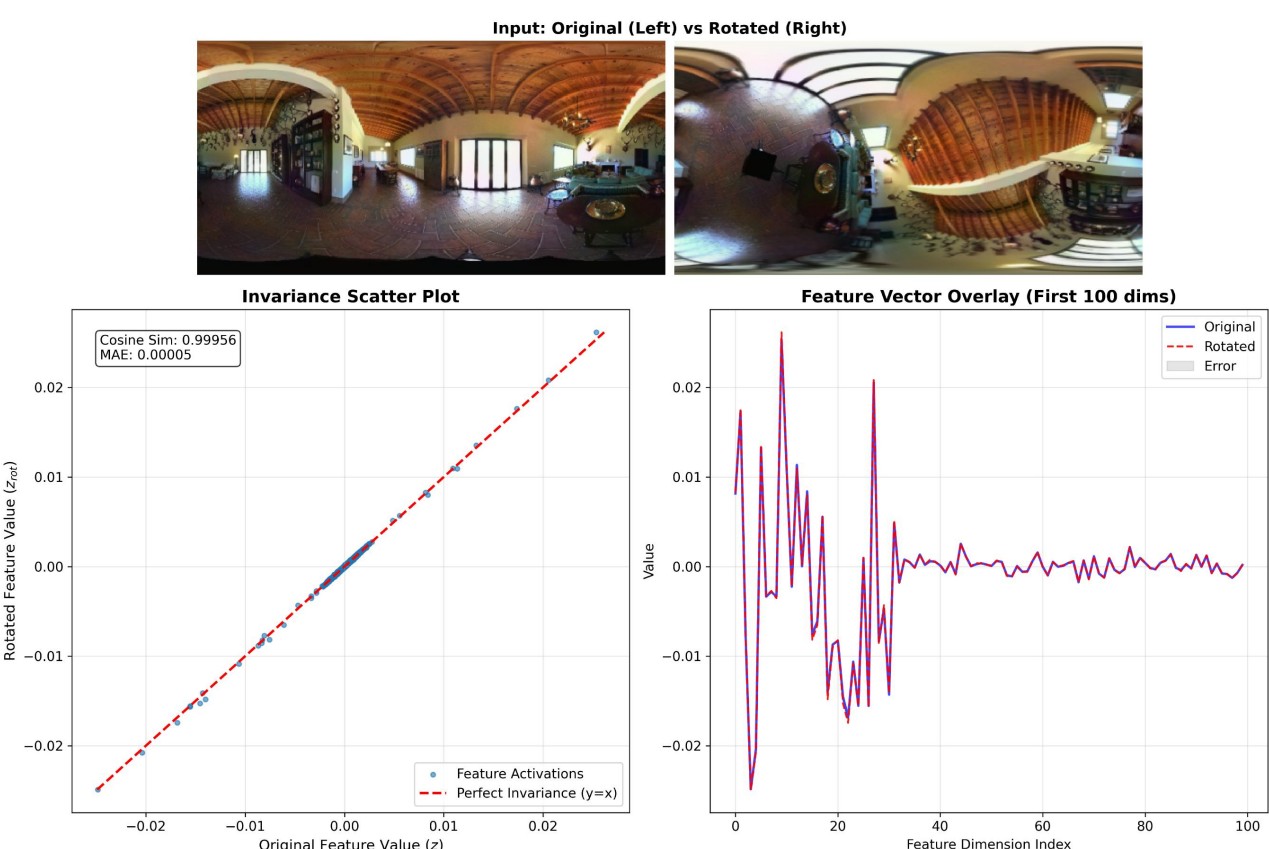

*Figure 7.* Analysis of Rotational Invariance in Feature Space. We evaluate the stability of the learned bispectrum features under arbitrary $SO(3)$ rotations. (Left) The Scatter Correlation plot compares feature activations of the original image ($x$-axis) versus the rotated image ($y$-axis). The tight clustering along the identity line ($y = x$) indicates high invariance. (Right) Feature Vector Overlay: A dimension-wise comparison where the blue solid line (Original Features) and the red dashed line (Rotated Features) exhibit near-perfect alignment. This visual overlap confirms that the bispectrum quantity effectively marginalizes geometric pose information and presents as a SO(3) invariant.

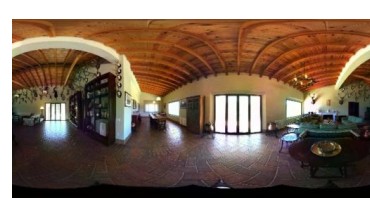

*Original Image*

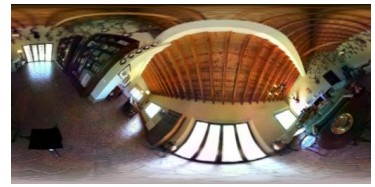

*Yaw:-156°,Pitch: -28°,Roll:36°*

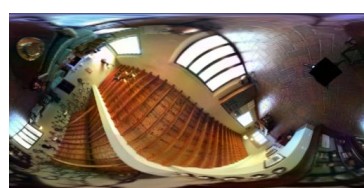

*Yaw:133°,Pitch: -1°,Roll:-121°*

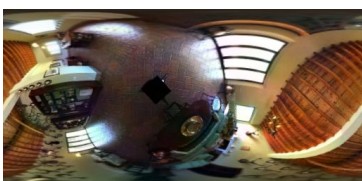

*Yaw:45°,Pitch: 81°,Roll:162°*

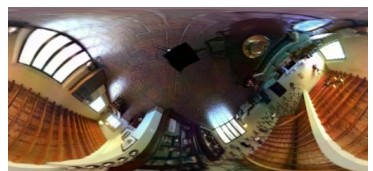

*Yaw:-79°,Pitch: 6°,Roll:-131°*

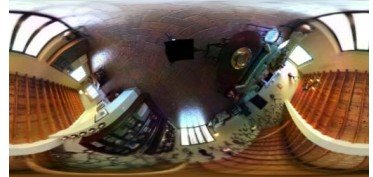

*Yaw:-57°,Pitch: 28°,Roll:-134°*

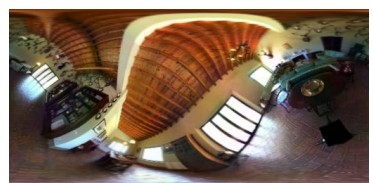

*Yaw:-174°,Pitch: -52°,Roll:-23°*

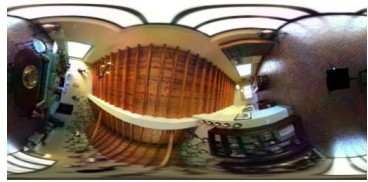

*Yaw:126°,Pitch: 37°,Roll:-103°*

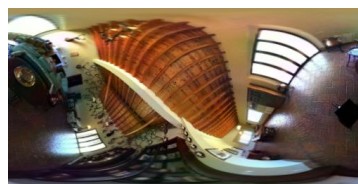

*Yaw:109°,Pitch: 0°,Roll:-75°*

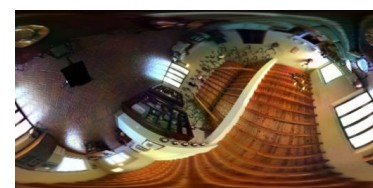

*Yaw:-22°,Pitch: -58°,Roll:-146°*

*Figure 8.* **Visualization of drastic geometric deformations induced by random** $SO(3)$ **rotations in the Equirectangular Projection (ERP) domain**. The top panel shows the original canonical view, while the subsequent panels display the same scene under randomly sampled 3D rotations (annotated with Yaw, Pitch, and Roll). Note that standard rigid 3D rotations manifest as complex, non-linear pixel displacements and distortions in the 2D ERP format. This visualization highlights the inherent challenge for baseline models to learn rotation robustness solely through data augmentation, motivating the need for our theoretically invariant architecture.

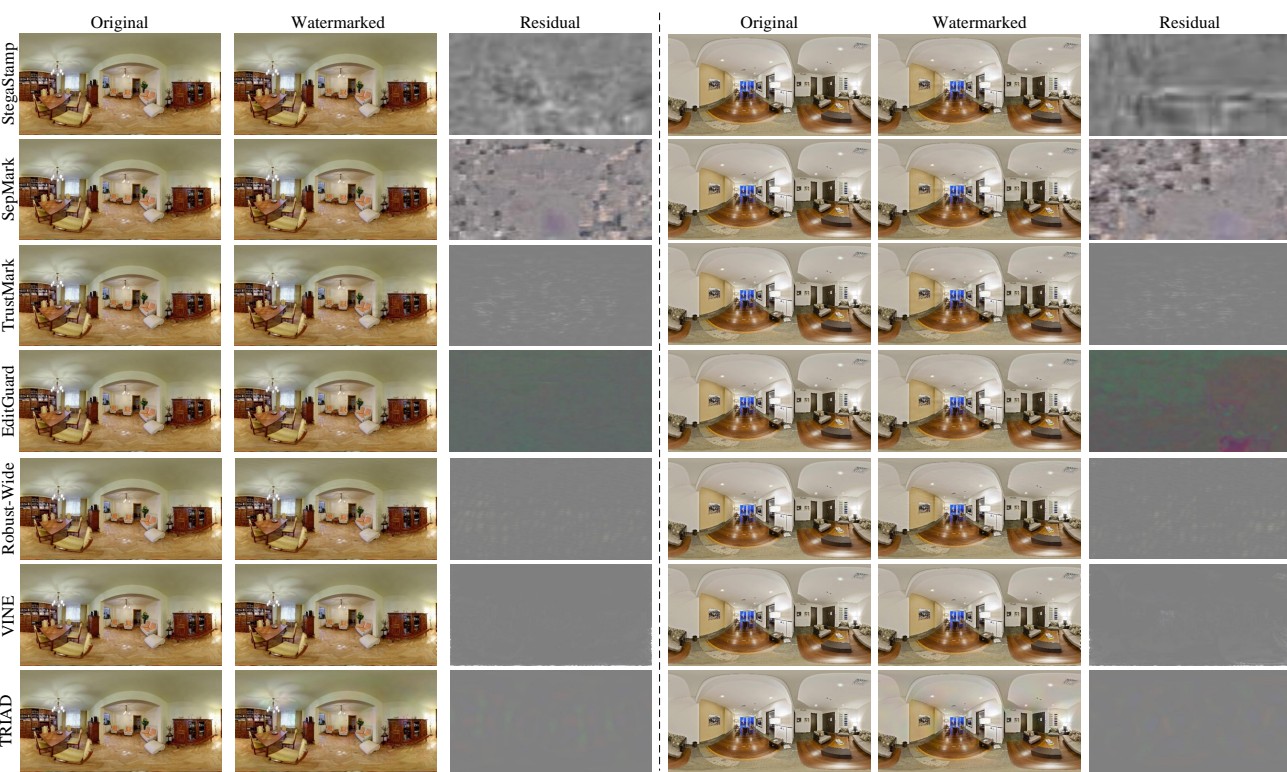

*Figure 9.* Qualitative evaluation of visual fidelity and imperceptibility. From left to right: the original cover panorama, the watermarked panorama, and the pixel-wise residual map.

