# OpenReview forum: "Rotation-Invariant Spherical Watermarking via Third-Order SO(3) Representation Coupling"
_ICML.cc/2026/Conference — ICML 2026 regular_

### Official Review · Reviewer_LqN5 · 2026-03-07

**Soundness:** 3
**Presentation:** 3
**Significance:** 3
**Originality:** 2
**Overall Recommendation:** 4
**Confidence:** 3

**Summary:**

This paper proposes TRIAD, a method designed to improve the robustness of watermarking in panoramic images under arbitrary 3D rotations. The approach models a panorama as a spherical signal and embeds the watermark into a subspace of high-order spherical harmonic coefficients in the spherical harmonic domain. On the decoding side, it constructs a rotation-invariant spherical bispectrum scalar by leveraging coupling and projection under the third-order SO(3) representation, enabling message recovery under unknown rotations. The paper provides the corresponding theoretical derivations and evaluates robustness against rotational attacks and a range of common distortions on the panoContext and SUN360 datasets, reporting results using metrics such as PSNR, SSIM, and Bit Accuracy.

Claims and Evidence

Claim 1 (Problem and methodological claim): The paper targets robust watermark extraction for panoramic images under arbitrary SO(3) rotations and proposes to replace pure data augmentation with spherical representations and invariant features. The problem formulation is clear, and the methodological motivation aligns well with the underlying spherical geometry. It would be helpful for the authors to further sharpen the boundary of the “core novelty,” clarifying which components constitute a standard spherical processing toolkit versus which design choices are genuinely novel and task-critical for watermarking.

Claim 2 (Theoretical claim on rotation invariance): The paper provides explicit definitions, constructions, and a derivation pathway for rotation invariance, and the appendix presents a proof framework consistent with the implementation. Overall, the theoretical chain is coherent. For claims related to “robustness,” the main text would benefit from a clearer distinction between conclusions that are strictly proven and those that are interpretive analyses grounded in structural properties, so that readers can correctly understand the scope of guarantees.

Claim 3 (Empirical results): The experiments generally support the claim that the method is highly robust to continuous rotations and maintains strong extraction accuracy under multiple distortions. The paper reports that TRIAD achieves 39.22 dB PSNR and 0.9946 SSIM at 32-bit payload, and presents Bit Accuracy under various distortions, along with evidence that the bispectrum is more stable than the power spectrum in high-payload settings. A more rigorous comparison under strictly matched payloads would strengthen the cross-method conclusions and avoid confounding effects from capacity differences.

Methods and Evaluation Criteria

Partially. The method is reasonably well-defined, with a clear end-to-end encoding and decoding mechanism, and the training objectives and evaluation metrics follow common practice for this task. For evaluation, two aspects could be strengthened: (i) more systematic comparisons under strictly aligned payloads, and (ii) more explicit reporting of end-to-end computational cost and deployment constraints to improve reproducibility and practical transferability.

Theoretical Claims

Partially checked. The main text (Theorem 4.1) and the appendix provide a formal construction and derivation framework for rotation invariance, with symbol definitions that correspond clearly to the implemented method. The paper would benefit from more precise statements in the main text regarding the objects and conditions covered by the theoretical results, especially by clarifying the relationship between “invariance” and claims such as “decodability” and “stable recovery,” so that readers do not conflate conclusions at different levels.

Experimental Designs or Analyses

The experimental design is well aligned with the paper’s central claims, including: (i) stress tests under random SO(3) rotations, (ii) common distortion tests such as JPEG, resizing, Gaussian noise, median filtering, brightness, and contrast, and (iii) ablations over V_embed, l_max, and bispectrum versus power spectrum. Further suggestions include: adding stronger payload-matched baseline comparisons, and incorporating test combinations that more closely reflect real-world content processing pipelines, to improve the external validity of the conclusions.

Supplementary Material

Read. I mainly referred to Appendix A.1 for proofs and supplementary experimental details.

Relation to Broader Scientific Literature

The paper connects naturally to prior work in spherical signal processing, SO(3) equivariant representation learning, higher-order invariants, and bispectrum-based techniques. The choice of problem setting and toolkit is internally consistent. The positioning could be strengthened by more explicitly articulating the paper’s incremental contribution relative to prior spherical representation and invariant-feature work, specifically in terms of the watermark embedding and recovery mechanism.

**Compliance With Llm Reviewing Policy:**

Affirmed.

**Final Justification:**

The rebuttal addressed several of my concrete concerns and strengthened my confidence in the paper’s technical soundness, particularly through the additional experiments and analysis. However, it did not fundamentally change my overall assessment regarding the paper’s contribution boundary, claim calibration, and evaluation scope, so I am maintaining my original Weak Accept recommendation.

**Key Questions For Authors:**

Major Comments

Improve readability and intuition. Add more intuitive explanations and figures in the methods section, clarifying how spherical harmonic coefficients transform under rotation, how bispectrum invariants remove rotational effects, and the intuition behind the embedding subspace choice.

Strengthen fairness via payload-matched comparisons. Add baseline comparisons under strictly aligned payloads, or explicitly foreground the impact of capacity differences in tables and conclusions.

Clarify the boundary of theoretical guarantees. More clearly distinguish the invariance theorem from other robustness arguments, including their conditions and evidential strength, so readers do not interpret interpretive analysis as a guarantee of the same level.

Add tests closer to real processing pipelines. Include more complex composite attacks or platform processing flows to strengthen evidence for generalization.

Minor Comments

Use terms such as “provable,” “guaranteed,” and “robust” more consistently, and ensure their scope and strength are clearly indicated.

Near results tables, more explicitly highlight the payload settings used by different methods to reduce the risk of misinterpretation.

Add a small number of failure or boundary cases to more clearly delineate the method’s applicability.

Questions for the Authors

Under strictly payload-matched conditions, does TRIAD’s advantage over each baseline remain consistent?

Over a broader range of V_embed and l_max, are there clear degradation regimes or numerical stability issues?

How do end-to-end inference latency and memory footprint scale at higher resolutions or higher payloads?

How robust is the method under more complex pipelines, such as editing followed by rotation, or reprojection combined with compression?

**Limitations:**

yes

**Strengths And Weaknesses:**

Strengths:

The task formulation is clear and practically meaningful, and focusing on panoramic rotation scenarios is well motivated.

The methodological narrative is coherent, with theory and experiments centered on the same core claim.

The rotation-robustness results are strong, and the ablation studies offer useful interpretability.

Weaknesses:

Claim strength versus evidence level could be aligned more rigorously. While the construction and proof chain for SO(3) rotation invariance is clear, the main-text phrasing may lead readers to equate “invariance” with “stable decodability” or “robustness to broader processing pipelines.” The paper should explicitly separate strictly provable statements from interpretive robustness arguments and align concluding language with the evidence level.

Fairness of horizontal comparisons remains debatable. When payload settings differ across methods, the trade-off between PSNR and Bit Accuracy should not be used to draw a strong claim of “overall superiority.” Payload-matched comparisons, or more explicit highlighting of capacity differences in tables and conclusions, would strengthen the fairness of the evaluation.

The evaluation setting may be somewhat idealized, limiting evidence for real-world generalization. The current tests focus on global SO(3) rotations of full spherical signals and common distortions. More realistic compound operations, such as reprojection plus cropping, or editing followed by rotation and compression, are not sufficiently covered. Since partial spherical signals are also acknowledged as an open issue, clearer experiments or failure-mode analyses would better support the stated boundary.

The contribution boundary could be sharper. Because the spherical representation and bispectrum invariants are closely tied to established toolchains, the paper may be perceived as strong engineering integration. The authors could more clearly isolate which design choices are indispensable for watermarking and support this with ablations or counterexamples demonstrating failure without those choices.

Reproducibility details and numerical stability discussion could be strengthened. Spherical harmonic truncation, embedding subspace selection, and implementation details can materially affect stability and reproducibility. More systematic implementation details and sensitivity analyses would be valuable.

---

> ### Author Rebuttal · Authors · 2026-03-30
>
> We sincerely appreciate your recognition of the practical significance of our problem formulation, the coherent alignment between our theory and experiments, and the strong empirical robustness and interpretability of our framework!
>
> ---
>
> **Question 1: Under strictly payload-matched conditions, does TRIAD’s advantage over each baseline remain consistent?**
>
> Thank you for this important question. We agree that payload-matched comparison is important for a fair assessment.
> Specifically, TrustMark and VINE use fixed 100-bit payload settings in their released checkpoints, which makes strict payload matching with other methods infeasible. In contrast, the released checkpoints of Robust-Wide and EditGuard use 64-bit payloads, which is the same as our 64-bit setting and therefore allow a more meaningful comparison.
> We conduct an additional comparison focusing on methods with comparable payload settings, and evaluate them under our rotation protocol in terms of both bit accuracy under arbitrary rotations and PSNR. The results show that TRIAD’s advantage remains consistent: it still achieves substantially stronger rotation robustness while maintaining competitive visual fidelity.
>
> |Method|Payload|PSNR|Bit Accuracy (Under 3D Rotations)|
> |-|-|-|-|
> |StegaStamp|64|28.91|0.503|
> |SepMark|64|34.18|0.518|
> |Robust-Wide|64|41.65|0.496|
> |EditGuard|64|36.58|0.487|
> |TRIAD (ours)|64|38.17|1.000|
>
>
> **Question 2: Over a broader range of V_embed and l_max, are there clear degradation regimes or numerical stability issues?**
>
> Thank you for this important question. We agree that the behavior over a broader range of $V\_{embed}$ and $l\_{max}$ should be examined more explicitly.
> To address this, we conduct additional ablations over a wider set of embedding subspaces and spectral cutoffs. The results are summarized below. Overall, we do not observe clear numerical instability in the tested range. Performance changes are generally smooth rather than abrupt, broader V_embed reduces PSNR, but incurs no obvious instability in robustness.
>
> |$l\_{max}$|$V\_{embed}$|PSNR|Bit Acc (Under 3D Rotations)|
> |-|-|-|-|
> |16|{6,8,14}|39.22|1.000|
> |20|{6,8,14,16}|39.16|1.000|
> |24|{6,8,14,16,20}|38.46|1.000|
> |28|{6,8,14,16,20,22}|37.19|1.000|
>
> **Question 3: How do end-to-end inference latency and memory footprint scale at higher resolutions or higher payloads?**
>
> Thanks for the question. The dominant cost in our model comes from the equivariant spectral blocks (e.g., tensor-product operations), whose complexity is determined by the chosen irreducible representations and is largely independent of image resolution or payload size.
>
> Increasing resolution mainly affects the grid transform and a lightweight image-space masking branch, while the core spectral tensor-product path remains unchanged. Increasing payload only enlarges small projection/readout MLPs, which contributes minimally to total time costs.
>
> As a result, both latency and memory grow only modestly with higher resolution or payload, as shown below:
> |Resolution|Payload|Inference Time(s)| Memory(GB)
> |-|-|-|-|
> |512x1024|32|0.0153|0.8|
> |512x1024|64|0.0179|0.8|
> |1024x2048|32|0.0296|1.2|
> |1024x2048|64|0.0318|1.2|
>
> **Question 4: How robust is the method under more complex pipelines, such as editing followed by rotation, or reprojection combined with compression?**
>
> Thank you for this important question. We conduct additional experiments under several compound attack pipelines that better reflect real-world usage. The results are summarized below:
> |Attack|Bit Accucary|
> |-|-|
> |Inpainting(5-10%) + Random 3D Rotation|0.925|
> |Crop(512x1024 ->480x960) + Resize(0.5x) |0.913|
> |Reprojection + JPEG(Q=60)|0.867|
> |Gaussian Blur (σ=3) + Random 3D Rotation|1.000|
>
> These results show that TRIAD remains highly robust under several non-ideal compound pipelines, especially when the full spherical structure is largely preserved (e.g., blur + rotation). As expected, the most challenging case is reprojection + compression, since reprojection changes the signal parameterization and may partially violate the complete spherical signal assumption underlying our theoretical guarantee.
> Overall, these experiments suggest that TRIAD’s robustness extends beyond idealized pure rotation settings, while also making clear its current boundary: the method is strongest when the spherical signal remains largely intact, and performance degrades more noticeably under operations that introduce partial-view / reprojection-induced information loss.

---

> > ### Author Rebuttal · Reviewer_LqN5 · 2026-04-03
> >
> > Thank you for the detailed rebuttal and the additional experiments. I will maintain my original score.

---

> > > ### Author Response · Authors · 2026-04-06
> > >
> > > Dear Reviewer LqN5,
> > >
> > > Thank you again for your valuable time and feedback.
> > >
> > > If helpful, we would be glad to provide further clarification on any aspect of our work. Please feel free to let us know if there are any additional questions or concerns during the discussion phase.
> > >
> > > Best regards,
> > >
> > > The Authors

---

### Official Review · Reviewer_YfSq · 2026-03-09

**Soundness:** 3
**Presentation:** 2
**Significance:** 2
**Originality:** 2
**Overall Recommendation:** 4
**Confidence:** 4

**Summary:**

This paper proposes a theoretically grounded framework for provably robust watermarking that delves into the natural spherical structure of panoramic images. By coupling higher-order irreducible representations via tensor products and projecting onto the trivial representation, it derives a spherical invariant bispectrum that preserves phase information while remaining strictly rotation-invariant. This allows watermarks to be embedded into higher-order spherical harmonic coefficients and reliably recovered from the bispectrum scalars. The proof and experiments demonstrate the robustness to continuous rotations and high visual fidelity.

**Compliance With Llm Reviewing Policy:**

Affirmed.

**Key Questions For Authors:**

1. Limited novelty of the invariant construction：The method relies on bispectrum invariants, which have been explored in prior work on invariant signal representations and have also been used previously in watermarking contexts (e.g., works combining Radon transforms and bispectrum invariants). While the paper adapts these ideas to the spherical signals, the manuscript could more clearly articulate what aspects of the invariant construction are fundamentally new.

2. Experimental comparisons could be broader：Although the experiments demonstrate robustness to rotations and certain perturbations, the set of baselines appears somewhat limited. Additional comparisons with other rotation-aware or spherical-domain watermarking approaches would help better position the contribution relative to existing work [1].
[1] Kong X. Rotation-Robust Deep Watermarking for Arbitrary Angular Transformations[C]//2025 2nd International Conference on Intelligent Perception and Pattern Recognition (IPPR). IEEE, 2025: 13-18.

3. Computational complexity is not fully discussed：The proposed approach relies on spherical harmonic decomposition and higher-order invariant construction, which may introduce nontrivial computational overhead. A discussion of computational cost and scalability would improve the practical assessment of the method.

**Limitations:**

yes

**Strengths And Weaknesses:**

Strengths
1. Timely and relevant problem：The paper studies watermarking for spherical images under arbitrary 3D rotations. This problem is increasingly relevant due to the growing use of panoramic visual data in many applications. Ensuring watermark robustness under SO(3) transformations is therefore a practically meaningful objective.

2. Strong theoretical for rotation-invariant：The method is grounded in spherical harmonic analysis and SO(3) representation theory, which provides a principled way to construct rotation-invariant watermark signals. Unlike many existing watermarking approaches that rely on data augmentation or heuristic robustness strategies, the proposed method explicitly models the SO(3) transformation group, which provides theoretical guarantees for rotation invariance.

3. Empirical validation：The experiments demonstrate that the proposed approach can maintain high watermark recovery accuracy under various rotations and distortions, suggesting that the method is effective in preserving watermark information under geometric transformations.

Weaknesses
1. Limited novelty of the invariant construction：The method relies on bispectrum invariants, which have been explored in prior work on invariant signal representations and have also been used previously in watermarking contexts (e.g., works combining Radon transforms and bispectrum invariants). While the paper adapts these ideas to the spherical signals, the manuscript could more clearly articulate what aspects of the invariant construction are fundamentally new.

2. Experimental comparisons could be broader：Although the experiments demonstrate robustness to rotations and certain perturbations, the set of baselines appears somewhat limited. Additional comparisons with other rotation-aware or spherical-domain watermarking approaches would help better position the contribution relative to existing work [1].
[1] Kong X. Rotation-Robust Deep Watermarking for Arbitrary Angular Transformations[C]//2025 2nd International Conference on Intelligent Perception and Pattern Recognition (IPPR). IEEE, 2025: 13-18.

3. Computational complexity is not fully discussed：The proposed approach relies on spherical harmonic decomposition and higher-order invariant construction, which may introduce nontrivial computational overhead. A discussion of computational cost and scalability would improve the practical assessment of the method.

4. Hard to Follow：The presentation of the methodology allows us to achieve the ideal. However, the text introduces several key concepts (such as spherical harmonics) using undefined acronyms and unexplained mathematical symbols. It is unclear what the abbreviation TRIAD stands for in the proposed framework.

---

> ### Author Rebuttal · Authors · 2026-03-30
>
> We thank the reviewer for recognizing the practical relevance of the problem, the principled theoretical foundation of our approach, and the strong empirical results demonstrating its effectiveness.
>
> ---
>
> **Question 1: Limited novelty of the invariant construction.**
>
> We thank the reviewer for this important comment. the key novelty of our work lies in its SO(3)-native formulation and how the invariant is constructed within the watermarking pipeline, which differs fundamentally from prior methods.
>
> Prior bispectrum-based watermarking methods use the bispectrum as a fixed invariant descriptor in transformed Euclidean domains. Both embedding and extraction operate entirely within the same fixed invariant feature space. In contrast, our construction is designed to solve a different problem: how to embed watermark information in non-invariant, higher-order spherical harmonic subspaces while still enabling strictly rotation-invariant extraction.
>
> This is the key novelty of our invariant construction. Specifically, we do not use the bispectrum as a handcrafted feature, but as a representation-theoretic bridge between equivariant embedding and invariant decoding. We embed watermark information into higher-order equivariant spherical harmonic subspaces that are rotation-sensitive and information-rich, and then recover it through a bispectrum-based invariant constructed via third-order coupling and projection onto the trivial representation, which creates an invariant scalar that still carries information from the non-invariant embedding space. This explicitly decouples embedding (in equivariant space) from extraction (in invariant space).
>
> Our invariant is not a generic descriptor, but a structured mapping that transports information from non-invariant coefficients to an invariant quantity, making recovery under arbitrary SO(3) rotations possible. To our knowledge, prior bispectrum-based watermarking methods do not address this setting, nor do they construct invariants for the purpose of enabling recoverability from equivariant embeddings.
>
> **Question 2: Experimental comparisons could be broader.**
>
> Thank you for the suggestion. We agree that broader comparisons can better position our method.
> Our original baselines are strong 2D robust image watermarking methods commonly designed for robustness against editing and traditional perturbations. However, 2D in-plane rotation and the 3D rotation of panoramic spherical signals are fundamentally different problems. A planar rotation is defined on a Euclidean image grid, while our setting considers a panorama as a signal on the sphere
> $S^2$, transformed by SO(3). Under ERP projection, an arbitrary 3D rotation does not correspond to a simple image-plane rotation, but induces global, latitude-dependent nonlinear distortions.
> The mentioned work [1] is relevant as a 2D rotation-aware watermarking method for planar images, but we were unable to include it directly because no official code are publicly available.
>
> To address this concern, we add another open-source 2D rotation-aware watermarking baseline[2] and evaluate it under our panoramic rotation protocol. The results show that although it improves over standard 2D baselines, it still degrades substantially under arbitrary 3D panoramic rotations (Yaw,Pitch,Roll) due to the induced  latitude-dependent distortions, while our method remains consistently robust due to its SO(3)-invariant spherical bispectrum design.
> |Method| (-59°, 28°, 100°) | (10°, 20°, -100°) | (-5°, 80°, 160°) |(9°, 15°, -178°) |(-5°, 80°, 160°) |(-5°, 80°, 160°) | Average |
> |-|-|-|-|-|-|-|-|
> |RoWSFormer[2]| 0.512 | 0.500 | 0.498 |0.476|0.469|0.500|0.493|
> |Ours|1.000|1.000|1.000|1.000|1.000|1.000|1.000|
>
> [2] Weitong Chen, Yuheng Li. "RoWSFormer: A Robust Watermarking Framework with Swin Transformer for Enhanced Geometric
> Attack Resilience." arXiv preprint arXiv:2409.14829 (2024).
>
> **Question 3: Computational complexity is not fully discussed.**
>
> Thank you for pointing this out.
> We note that a detailed computational complexity and runtime analysis is already included in the Appendix C.1 and we also provide the results below. The main additional cost of our method comes from the spherical harmonic transform (SHT/ISHT) and the equivariant spectral blocks. TRIAD demonstrates a comparatively low cost both in inference time and in GPU usage.
>
> |Method|EncodingTime(s)|DecodingTime(s)|TotalTime(s)|Memory(GB)|
> |-|-|-|-|-|
> |StegaStamp|0.0354|0.0318|0.0672|2.0|
> |SepMark|0.0053|0.0055|0.0108|0.9|
> |TrustMark|0.0197|0.0147|0.0344|0.6|
> |EditGuard|0.1659|0.0773|0.2432|1.7|
> |Robust-Wide|0.0111|0.0157|0.0268|3.0|
> |VINE|0.0583|0.0103|0.0686|4.9|
> |TRIAD(Ours)|0.0081|0.0072|0.0153|0.8|

---

> > ### Author Rebuttal · Reviewer_YfSq · 2026-04-02
> >
> > I would like to thank the authors for their response. I am keeping my current score.

---

> > > ### Author Response · Authors · 2026-04-06
> > >
> > > Dear Reviewer YfSq,
> > >
> > > Thank you for dedicating your time and effort to reviewing our paper!
> > >
> > > If you have any further concerns, please do not hesitate to let us know. We would be happy to provide any additional explanations that may be helpful during the discussion phase.
> > >
> > > Best regards,
> > >
> > > The Authors

---

### Official Review · Reviewer_tfkE · 2026-03-13

**Soundness:** 3
**Presentation:** 1
**Significance:** 2
**Originality:** 3
**Overall Recommendation:** 4
**Confidence:** 3

**Summary:**

This paper explores watermarking for spherical images.  The fundamental challenge for watermarking is to introduce a signal that can still be extracted after transformations, while simultaneously being visually imperceptible. The challenge for spherical images in particular is in dealing with the specific transformation of 3D rotations. This paper introduces TRIAD, which embeds watermarks in higher-order spherical harmonic coefficients. The watermark can be recovered from third-order coupling of SO(3) irreducible representation features projected to rotation-invariant scalars.

**Compliance With Llm Reviewing Policy:**

Affirmed.

**Final Justification:**

The rebuttal addressed some of my questions. However, a number of my concerns stem from the overall unclear writing throughout the paper.  While this was acknowledged in the rebuttal, it would constitute a major revision of the paper, and thus without the benefit of seeing the revision I will maintain my original rating.

In my initial review I also mentioned that I am not familiar with the watermarking literature, and so cannot comment confidently on the experimental analysis. It seems reviewer *YfSq* has more familiarity with this application domain, and they raised some related points in their review.  I defer to their opinion on the status of experimental comparisons.

**Key Questions For Authors:**

See questions in the Strengths and Weaknesses section. Addition question below:

- I’m curious how Figure 6 in the Appendix shows no change in energy in the degrees other than the embedding degrees.  I understand the embedding is restricted to V_embed, but in practice the watermark signal gets multiplied by learnable mask and is added to the original signal, so shouldn’t we expect *some* changes across all degrees?

**Limitations:**

yes

**Strengths And Weaknesses:**

**Strengths**
- This paper is the first to investigate the unique challenge of watermarking panoramic images.
- The proposed technical solution is well-motivated and intuitive. A solution based on rotation-equivariant models and rotation-invariant scalar projection seems like a great way to tackle this problem.
- As this is the first rotation-equivariant/invariant watermarking scheme for spherical images, the results are superior to traditional watermarking tools. This method would provide a reasonable baseline for future works in this area.

**Weaknesses**
- While the method is conceptually intuitive, the technical details behind the method would be difficult to follow for non-experts. Improving the writing would resolve much of the difficulty.
    - For example, the key idea behind the method is the bispectrum invariant defined in eq 6. The text “Concretely, the resulting bispectrum invariant I is defined as: … C^{0,0}_{l1,m1,l2,m2,l3,m3} denotes the Clebsch–Gordan coefficients corresponding to projection onto the trivial representation.”  Given the importance of this equation, why not define how the Clebsch-Gordan coefficients are computed, or why not be explicit about what “projection onto the trivial representation” means?  These points are representative of the overall issues with the presentation of this paper.
    - The details become clear for readers who are deeply familiar with the e3nn paper. Perhaps a simple solution is to refer readers to that source for background.

- Some writing could be made more precise in the Appendix.
    - “implying stability of the bispectrum invariant under isotropic blur.” If I understand correctly, what equation 17 shows is that the bispectrum scalar undergoes a complex scaling under blur, so I’m not sure why this implies “stability.”
    - “As long as a sufficient subset of frequency triplets remains, the invariant does not collapse…” Again, making the language more precise would help. Explaining what is meant by “not collapse” would be helpful.  Maybe there is more that can be said about the power spectrum of natural images and losing only the (lower power) higher frequency components.
    - The distortions studied in A.2 could also be quantitatively analyzed in isolation. The evaluations in the main paper provide evidence of stability but evaluation of the entire model conflates many different factors.

- The choice of L_embed = {6,8,14} seems arbitrary, and potentially the optimal choice would depend also on the spectral properties of the dataset.  This hardcoded choice may not generalize well.

- Related work suggestion: "Spin-Weighted Spherical CNNs", Esteves et al, 2020.

*I'm not well versed in the watermarking literature, so I cannot confidently assess this paper's experimental analysis.  I look forward to discussing this with the other reviewers.*


Minor typos:
- Page 6: “Tthe”
- Figure 2: “Watermared”

---

> ### Author Rebuttal · Authors · 2026-03-31
>
> We sincerely appreciate your recognition of our focus on the important and underexplored area of panoramic watermarking robustness against SO(3), the well-motivated intuition, the novelty and effectiveness of our method!
>
> ---
>
> **Question 1: Given the importance of equation 6 , why not define how the Clebsch-Gordan coefficients are computed and “projection onto the trivial representation”?**
>
> Thank you for your question. We agree that Eq.(6) is central to our method and should be explained more explicitly. The Coupling coefficients used in Eq. (6) can be computed via Wigner 3-j symbols as
>
> $
> C\^{0,0}\_{l\_1 m\_1 l\_2 m\_2 l\_3 m\_3} =
> \sqrt{\frac{(2l\_1+1)(2l\_2+1)(2l\_3+1)}{4\pi}}
> \begin{pmatrix}
> l\_1 & l\_2 & l\_3 \\\\
> 0 & 0 & 0
> \end{pmatrix}
> \begin{pmatrix}
> l\_1 & l\_2 & l\_3 \\\\
> m\_1 & m\_2 & m\_3
> \end{pmatrix}.
> $
>
> Here, “projection onto the trivial representation” means coupling
> $V_{l\_1}\otimes V_{l\_2}\otimes V_{l\_3} \rightarrow V\_0$
> i.e., the total angular momentum is zero.
>
> **Question 2: Precise terminology ("stability", "not collapse") in the Appendix**
>
> We thank the reviewer for pointing out the imprecise wording. We will revise this part for clarity.
>
> (1) Our notion of "stability" here refers to the continuity of the bispectrum with respect to perturbations induced by blur.
> Eq. (17) shows that isotropic Gaussian blur rescales spherical harmonic coefficients as
> $\tilde c\_l^m = g(l) c\_l^m, \quad g(l) = e^{-\sigma^2 l(l+1)},$ where $g(l)$ is a smooth, bounded function with $0 < g(l) \le 1$. Substituting into the bispectrum definition yields
> $\tilde I = \sum g(l_1) g(l_2) g(l_3) \cdot (\cdots),$
> so the bispectrum is smoothly attenuated, not arbitrarily distorted, as blur increases.
> For fixed bandwidth
> $l≤l\_{max}$​, the perturbation remains bounded, i.e. $|\tilde I -I|$  is controlled by the deviation of g(l) from 1. We will replace “stability” with a more precise statement.
>
> (2) We agree "not collapse" is informal. What we mean is that the invariant representation remains non-degenerate / informative under moderate low-pass filtering, rather than becoming nearly constant across inputs.
> Under anti-aliasing or resizing, high-frequency coefficients are attenuated or truncated, so the bispectrum is computed from a reduced set of frequency triplets. However, for natural images, most spectral energy is concentrated in low-to-mid frequencies, while the highest frequencies typically carry less energy. Hence moderate low-pass filtering mainly removes weaker high-frequency interactions, while many informative low-/mid-frequency bispectral couplings remain.
> We will replace “does not collapse” with a more precise statement under realistic spectral assumptions.
>
> **Question 3: The distortions studied in A.2 could also be quantitatively analyzed in isolation.**
>
> We provide the impacts of distortions on bispetrum in isolation below. Cosine similarity is calculated between the bispectrum before and after distortions. The results validates the stability analyzed in A.2.
> |Distortion|Strength|Cosine Similarity|
> |-|-|-|
> |Gaussian blur| σ=3 |0.997849 |
> |Resize|0.5x|0.999972|
> |Gaussian noise|deviation=0.05|0.999965|
> |Combined|/|0.997458|
>
> **Question 4: The choice of $L\_{embed}.$**
>
> We thank the reviewer for this comment. The choice of $L\_{embed}$ is not intended as a dataset-specific tuned optimum, but as a principled multi-band configuration. In particular, it is structurally compatible with the bispectrum coupling in our decoder: under Clebsch–Gordan rules, information from $V_6$ and $V\_8$ can naturally couple into $V\_{14}$ since $6+8=14$, which facilitates informative third-order invariant extraction.
>
> Importantly, the method is not sensitive to this exact choice. As shown below, alternative configurations such as {5,9,13} and {6,8,10,14} yield very similar performance, indicating that the framework generalizes across reasonable multi-band selections rather than depending on a single hardcoded setting. We use {6,8,14} as the default because it achieves comparable performance with a simpler configuration.
>
> |$L\_{embed}$|PSNR|SSIM|Bit Acc|
> |-|-|-|-|
> |{5,9,13}|39.15|0.9937|1.000|
> |{6,8,10,14}|38.97|0.9913|1.000|
> |{4,8,11,14}|38.73|0.9907|0.998|
> |{6,8,14}(ours)|39.22|0.9946|1.000|
>
> **Question 5: why is no change in energy in the degrees other than the embedding degrees?**
>
> In principle, spatial masking and reconstruction could introduce some cross-degree leakage. However, in practice this leakage is negligible. The watermark is injected only into $V\_{embed}$, and the decoder is trained to recover information specifically from invariant features derived from these subspaces. Changes outside $V\_{embed}$ therefore do not improve decoding, but they do increase distortion and are penalized by the fidelity loss. As a result, training naturally concentrates the watermark energy in $V\_{embed}$ and suppresses off-subspace changes. Figure 6 reflects this learned near-isolation rather than a strict analytical guarantee.

---

> > ### Author Rebuttal · Reviewer_tfkE · 2026-04-03
> >
> > I appreciate the author responses to my questions.  Between these responses, and the other reviews+rebuttals, I have enough information to finalize my ratings.

---

> > > ### Author Response · Authors · 2026-04-06
> > >
> > > Dear Reviewer tfkE,
> > >
> > > We sincerely appreciate the time and effort you have dedicated to reviewing our paper.
> > >
> > > Thank you very much!
> > >
> > > Best regards,
> > >
> > > The Authors

---

### Decision · Program_Chairs · 2026-04-30

**Decision:**

Accept (regular)

**Comment:**

This submission studies a timely and meaningful problem and proposes a technically grounded approach to watermarking panoramic images under arbitrary 3D rotations. Across the reviews, there is broad agreement that the paper is sound, well motivated, and empirically strong on the central rotation robustness claim, with especially positive feedback on the SO(3) formulation and the use of invariant bispectral features. The main reservations are about clarity of presentation, sharper positioning of the novelty relative to prior invariant and bispectrum work, and the need for broader and more carefully matched evaluations. The rebuttal addressed several concrete questions and improved confidence, but it did not fully remove concerns about exposition and evaluation scope.